# A Computational Theory for Efficient Mini Agent Evaluation with Causal Guarantees

## Abstract

In order to reduce the cost of experimental evaluation for agents, we introduce a computational theory of evaluation for mini agents: build evaluation model to accelerate the evaluation procedures. We prove upper bounds of generalized error and generalized causal effect error of given evaluation models for infinite agents. We also prove efficiency, and consistency to estimated causal effect from deployed agents to evaluation metric by prediction. To learn evaluation models, we propose a meta-learner to handle heterogeneous agents space problem. Comparing with existed evaluation approaches, our (conditional) evaluation model reduced 24.1% to 99.0% evaluation errors across 12 scenes, including individual medicine, scientific simulation, social experiment, business activity, and quantum trade. The evaluation time is reduced 3 to 7 order of magnitude per subject comparing with experiments or simulations.

## 1 Introduction

Throughout the history of computer science, manually handing the data has been replaced by computational approach, such as computational linguistic (OpenAI et al. [2024], DeepSeek-AI et al. [2025]), computational biology (Abramson et al. [2024]), and computational learning (Valiant [1984]). Here we extend the computational theory to the ubiquitous domain of evaluation.

Randomized controlled trials (Fisher [1974], Box et al. [2005]) are widely regarded as the gold standard for evaluating the efficacy of interventions, such as drugs and therapies, due to their ability to minimize bias and establish causal relationships between interventions and outcomes. By randomly assigning participants to intervention group or control group, RCTs effectively balance both observed and unobserved confounding variables, thereby mitigating the influence of hidden common causes (Reichenbach [1999]). However, conducting RCTs to assess every potential agent is often prohibitively expensive and, in certain cases, unfeasible as shown in appendix A. This challenge is particularly pronounced in fields like artificial intelligences, where models may possess hundreds of parameters, resulting in an infinite and high-dimensional space of potential interventions. Evaluating such agents through randomized experiments is resource-intensive, demanding substantial investments of time, finances, personnel, and materials (Speich et al. [2018]). Moreover, the extended duration of these experiments hampers the rapid iteration and optimization of agents, rendering the process inefficient. Given these limitations, alternative methodologies that balance rigorous evaluation with practical feasibility are essential, especially in rapidly evolving domains requiring swift assessment and deployment of new agents.

An improvement is to build the centralized A/B test platform, which has been proven successes in application updates and recommendation in Byte Dance (Byte-Dance). The implementation of centralized A/B testing platforms can significantly reduce the costs associated with conducting randomized trials for researchers. However, this cost reduction does not inherently enhance the utility

derived from each individual trial and it still can not handle the infinite evaluation subjects problem. Notably, the lower cost and increased accessibility of such platforms can lead to a substantial rise in the number of randomized trials conducted. This proliferation of experiments raises ethical concerns regarding participant exposure to potential harm during the randomization process. For instance, as highlighted by Zhou et al. [2024b], the automation and scalability of A/B testing frameworks, while beneficial for rapid experimentation, necessitate careful consideration of ethical implications to safeguard participant well-being. Polonioli et al. [2023] emphasizes the need for ethical and responsible experimentation to protect users and society.

In addition to other agent evaluation works in appendix B, evaluatology (Zhan et al. [2024]) emerges as an independent filed in recent years. However, there is still a lack of a theory to help us find an effective and efficient evaluation system that satisfies the concerns and interests of the stakeholders. Here, we introduce a computational theory to evaluate the effect of given agents. The main benefit to evaluate by computation is its efficiency and low cost. If we learned the connection between the experiment result and pre-experiment agents by a computational model, the evaluation cost will be reduced dramatically comparing with performing experiments.

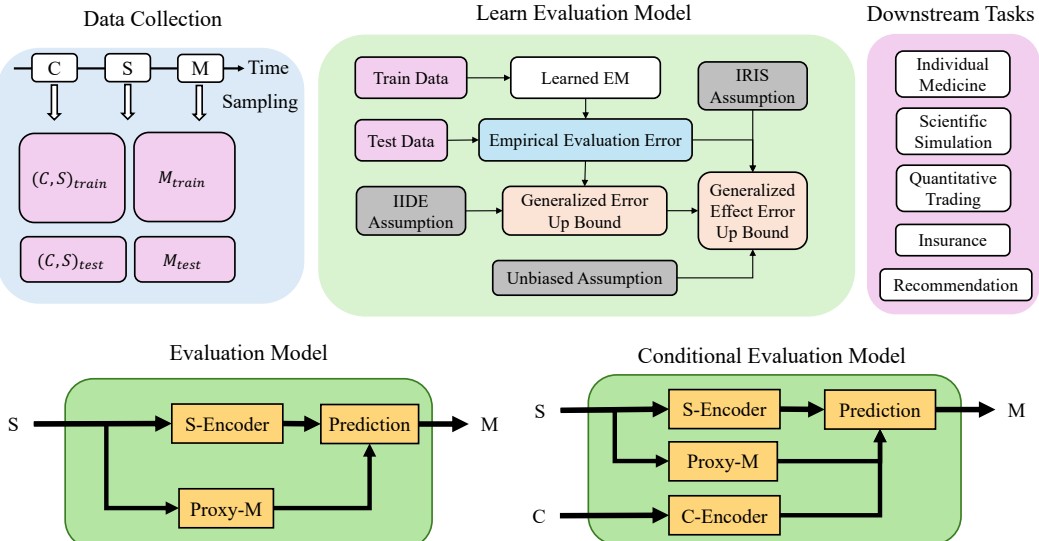

Figure 1: Procedure of computational evaluation. C is evaluation condition, S is evaluation subject (agent), M is evaluation metric, EM is evaluation model.

The procedure of our computational evaluation is listed in figure 7. First, data of evaluation condition, deployed agents, and evaluation metric are collected. If causal relationship is needed, the subject is required to be independently, randomly, identically sampled (IRIS) so that the causal effect from the subject to the metric can be bounded in the following steps. Next, evaluation model is learned from the collected data, and generalized evaluation error is upper bounded under independently, identically distributed error (IIDE) assumption. Also, the evaluation model is assumed unbiased to bound the causal effect from agent to metric. Then, learned evaluation model is applied to the downstream applications once the upper bounds are low enough to satisfy the real need. The mentioned assumptions are assessed by testing their necessary conditions.

In addressing on on the heterogeneous agents space problem in evaluation model learning, we propose a meta algorithm to parameterize agents, learn (conditional) evaluation models, and inference separately. The upper bounds of generalized error for both metric and its first order difference on agents are minimized by minimizing the empirical error. Also, learning evaluation model from evaluation samples directly is very challenging in some cases. So we add a proxy module for the agents in addressing the challenge. The proxy module will create some proxy metrics according some existing computational evaluation methods, which can help us to improve the performance of evaluation model.

In our evaluation experiment, we test evaluation model on 11 scenes including individual medicine, scientific simulation, insurance, advertisement, trade. Comparing with the existed computational

evaluation methods, our method reduced 24.1% to 99.0% evaluation errors while remain the advantage of computational evaluation efficiency. We also test our conditional evaluation model in the quantitative trade scene. Comparing with baseline backtesting strategy, our evaluation model reduced 89.4% evaluation errors. The evaluation acceleration ratio is ranged from 1000 times to 10000000 times comparing with experimental or simulation based evaluation in the 12 scenes.

## 2 Notation

| Variable | Name | Description |
|---|---|---|
| $S$ | evaluation subject (agents) | the operable and indivisible unit that we want to deploy, remove, understand, and modify |
| $C$ | evaluation condition | set of pre-evaluation features of evaluation subject, including inner attributes and environments that identify a real evaluation process |
| $M$ | evaluation metric | subset of stakeholders' buy-in metrics |
| $EM$ or $\hat{f}$ | evaluation model | a function that takes an evaluation subject and evaluation conditions as inputs and produces evaluation metrics as its output |
| L | evaluation error function (ranged from 0 to 1) | a function to measure the discrepancy between output of evaluation model and true metric |
| $E_{emp}$ | empirical evaluation error | average evaluation error of empirical sample |
| $E_{gen}$ | generalized evaluation error | expected evaluation error of super-population |

Table 1: Notation table for used variables.

Table 1 shows the notation we used in this paper. In order to simplify the evaluation problem, evaluation metric and difference of evaluation metric are assumed as measurable and pairwise comparable. The computational evaluation problem is modeled as computing the evaluation metrics for evaluation subject with certain evaluation conditions. Unless otherwise specified, all evaluation subjects in this paper are mini agents.

## 3 Theoretical analysis

### 3.1 Upper bound of generalized evaluation error

**Theorem 1.** ***Upper bound***. *Given any evaluation model $\hat{f}$, $P(E_{gen}(\hat{f}) < E_{emp}(\hat{f}) + \sqrt{\frac{1}{2n}\ln(\frac{1}{\sigma})}) \geq 1 - \sigma$ where $n$ is number of independent identical distributed error (IIDE) measurements, $0 < 1 - \sigma < 1$ is confidence.*

In Theorem 1, we show that the generalization error $E_{gen}$ is bounded by the empirical error $E_{emp}$. the number of error measurements $n$, and the significance level $1 - \sigma$ under the IIDE assumption. This result underpins our ability to use a finite number of evaluation samples to compare the performance of evaluation models across an infinite subject and condition space. For example, when the evaluation subject space is extremely large or infinite—as in the cases of AI diagnosis agents, AI treatment agents, or quantitative strategy agents—Theorem 1 allows us to generalize our empirical analysis to the entire subject space. Similarly, in evaluation condition spaces, such as those involving AI-driven discovery of unseen materials based on molecular and structural conditions, the generalization errors for any evaluation model can be bounded using the same theorem.

Once the upper bound is smaller than given error tolerance, evaluation model $f$ is regarded to meet the practical needs with high probability $1 - \sigma$. Additionally, Theorem 1 provides a method to estimate the number of error measurements (or evaluation samples) required prior to sampling.

We emphasize that IIDE can also be satisfied if the non-IID components of $(C, S, M)$ was offset by the evaluation model or the evaluation error function (such as non-misspecified time-series model on time-series data) although independent identical distributed $(C, S, M)$ is a sufficient condition of IIDE. In fact, the assumption of independent and identically distributed samples is considered excessively strong in this paper, particularly with respect to IID conditions and IID outcomes.

## 3.2 Hidden common cause

The hidden common cause between subject and metric may introduce the confounding bias if the evaluation task is causality sensitive, such as individual medicine. Causal inference is often used to handle hidden common cause from data. However, applying existed methods to handle unmeasured hidden common cause problem in evaluation faced those challenges:

- **Hidden identical common cause (HIC)**. If the assignment of $S$ is determined by an unlisted identical common cause $E$ which can be denoted as $S := E$, and there exists effect from $E$ to $M$. We may mistakenly attributing the effect on $M$ of $E$ to $S$. The identical hidden confounder can *never* be excluded from statistical analysis due to the identical property if $S$ was never controlled by randomization. Therefore, effect from $S$ to $M$ is unidentifiable in all kinds of existed non-intervention based causal analysis, including diagram-based identification (Pearl [1995], Tian and Pearl [2002], Shpitser and Pearl [2006], Bareinboim and Pearl [2012], YAN [2025]), neural-based identification (Xia et al. [2021]), statistical-based identification (Jaber et al. [2019]), observation study (ROSENBAUM and RUBIN [1983]), causal representation learning (Schölkopf et al. [2021]), and causal discovery (Spirtes and Glymour [1991], Zhang and Hyvärinen [2009]).

- **High-dimensional evaluation subject**. Potential outcomes (Splawa-Neyman et al. [1990], Holland [1986], Imbens and Rubin [2015]) often assumes each evaluation subject must be sampled once which often do not established for high-dimensional evaluation subjects. For instance, parameters of AI model can be hundreds or thousands and more complex which structure is different from continuous dose-level analysis (Schwab et al. [2020], Nie et al. [2021], Zhu et al. [2024], Nagalapatti et al. [2024]) with low dimension.

For causality sensitive task, any hidden common cause between subject and metric is excluded including HIC if the subject was randomly, independently, identically sampled or assigned as shown in theorem 2 and theorem 4. For high-dimensional evaluation subject, the causal effect of evaluation error is bounded and evaluation result can be generalized to the whole subject space whatever how large the space size as shown in theorem 4.

**Theorem 2.** *Strict causal advantage*. *Given $\{c_i, s_i, m_i\}_{i=1}^n$, where $c$ is evaluation condition, $s$ is independent from $c$, and $e$, and $s$ is independently, randomly, identically sampled (IRIS) from distribution $Pr(S)$, $m$ is generate by an unknown and inaccessible function $m = f(c, e, s)$, $e$ is other unlisted random variable, evaluation error function $L$ is mean square error, $\forall \hat{f}_1, \forall \hat{f}_2$ where $\hat{f}_1$ and $\hat{f}_2$ are unbiased estimation of $f$ given $c$ (CU), we have **Efficiency**: if $E_{gen}(\hat{f}_1) < E_{gen}(\hat{f}_2)$ for any $s$, then $\forall s_a \in \mathbf{S}, \forall s_b \in \mathbf{S}, \forall c \in \mathbf{C}, E_{gen}(\hat{f}_1(c, s_a) - \hat{f}_1(c, s_b)) < E_{gen}(\hat{f}_2(c, s_a) - \hat{f}_2(c, s_b))$; **Consistency**: if $\lim_{\hat{f} \to f} E_{gen}(\hat{f}) = 0$, then $\forall s_a \in \mathbf{S}, \forall s_b \in \mathbf{S}, \forall c \in \mathbf{C}, \lim_{\hat{f} \to f} E_{gen}(\hat{f}(c, s_a) - \hat{f}(c, s_b)) = 0$.*

From theorem 2, minimization of generalized mean square causal effect evaluation error can be performed by minimizing generalized evaluation error greedily until the evaluation error is close to 0. However, we can not get the generalized evaluation errors in reality. An upper bound with empirical errors was inferred in the theorem 3.

**Theorem 3.** *Causal bound with positivity*. $\forall s_a \in S, \forall s_b \in S$, *given any unbiased evaluation model* $\hat{f}(c, s_a)$ *and* $\hat{f}(c, s_b)$,

$$P(E_{gen}(\Delta \hat{f}) < 2 \max\{E_{emp}(a) + \sqrt{\frac{1}{2n_a} \ln(\frac{2}{\sigma})}), E_{emp}(b) + \sqrt{\frac{1}{2n_b} \ln(\frac{2}{\sigma})}\}) \geq 1 - \sigma$$

, *where* $\Delta \hat{f} = \hat{f}(c, s_a) - \hat{f}(c, s_b)$, $n_a$ *and* $n_b$ *are number of independently, randomly, identically sampled error measurements (IIDE) where $s = s_a$ and $s = s_b$, $0 < 1 - \sigma < 1$ is confidence, evaluation error function $L$ is mean square error ranged from 0 to 1.*

In practice, $n_a$ and $n_b$ are typically zero for infinite evaluation subject (non-positivity Imbens and Rubin [2015]); in such cases, upper bound of generalized effect error is given by Theorem 4.

**Theorem 4.** *Causal bound with non-positivity*. *Given unbiased evaluation model* $\hat{f}$, *then* $\forall s_a \in S, \forall s_b \in S$,

$$P(E_{gen}(\Delta \hat{f}) < 2(E_{emp}(\hat{f}) + \sqrt{\frac{1}{2n} \ln(\frac{1}{\sigma}))}) \geq 1 - \sigma$$

152 , *where $\Delta \hat{f} = \hat{f}(c, s_a) - \hat{f}(c, s_b)$, $n$ is number of independently, randomly, identically sampled error*
153 *measurements (IIDE), $0 < 1 - \sigma < 1$ is confidence, $s$ is independently, randomly, identically sampled*
154 *(IRIS), evaluation error function $L$ is mean square error ranged from 0 to 1.*

155 Theorem 4 provides the rationale for why our evaluation models can learn the causal effect from
156 subjects to metrics rather than mere correlations when the evaluation subjects are independently,
157 randomly, and identically sampled (IRIS), the evaluation models are unbiased, and the evaluation
158 errors are independently, randomly, identically distributed (IIDE). Also, effect error is upper bounded
159 by its bias even the model is not unbiased. Proofs of all the theorems were given in appendix C.

160 Comparing with Saito and Yasui [2020] and Alaa and Van Der Schaar [2019], our unbiased assumption
161 has potential to be rejected from real data testing as shown in the experiment part while they did
162 not provide approaches to reject their unbiased causal effect assumption (Saito and Yasui [2020])
163 or min-max optimal effect estimator assumption (Alaa and Van Der Schaar [2019]) from real data
164 without counterfactual. Also, IID samples assumption in those works (Saito and Yasui [2020],
165 Johansson et al. [2022], Alaa and Van Der Schaar [2019], Shalit et al. [2017], Li and Pearl [2022],
166 Wang et al. [2022]) is too strong for causal evaluation, especially IID conditions and IID outcomes.

# 4 Learning evaluation models

168 In evaluation processes, two critical factors are evaluation error and evaluation cost. Our objective is
169 to reduce evaluation cost while maintaining an acceptable level of generalized evaluation (causal)
170 error through computational evaluation. To achieve this, we can minimize empirical evaluation error
171 because upper bound size is fixed given number of error measurements $n$ and confidence requirement
172 $1 - \sigma$ in theorem 1 and theorem 4.

173 Directly learning an evaluation model from evaluation samples presents significant challenges. To
174 address this, we introduce a proxy module that generates surrogate metrics using established compu-
175 tational methods, such as dataset-based approaches including bootstrapping, hold-out validation, and
176 cross-validation. These techniques enhance the performance of the evaluation model by providing
177 more robust and reliable features. Furthermore, to effectively manage heterogeneous agents, we
178 employ vectorization strategies, develop specialized sub-models, and conduct inferences tailored
179 to each agent type. This approach ensures that the evaluation model can accommodate the diverse
180 characteristics inherent in heterogeneous agent spaces.

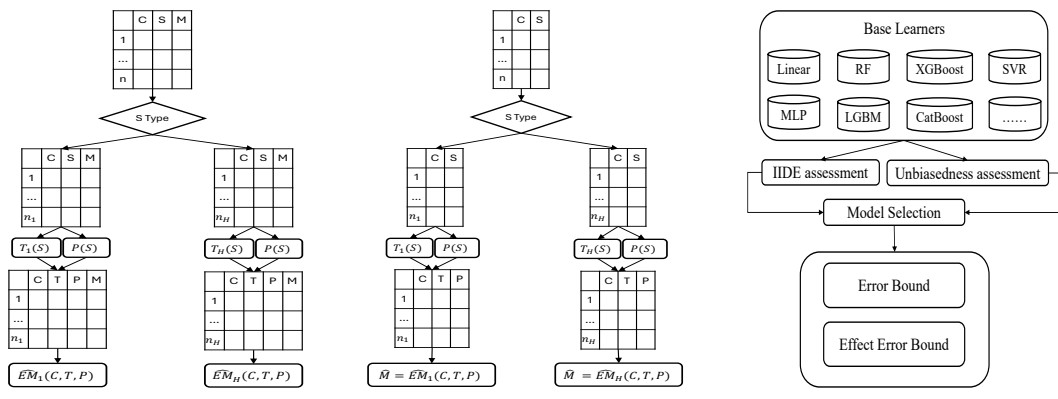

(a) Evaluation model learning.   (b) Evaluation model inference.   (c) Evaluation model Selection.

Figure 2: Learn evaluation models from data. $P$ is proxy metrics of evaluation subject, and $T_i$ is
tensorization function for subject type $i$.

181 The proposed meta-learning algorithm for evaluation model is illustrated in Figure 2. Initially,
182 data are classified based on subject type. Subsequently, distinct vectorization methods are applied
183 corresponding to each subject type. Proxy metrics are then computed using predefined proxy
184 functions, facilitating the development of tailored evaluation models. For the assessment of unseen
185 subjects, the process involves computing proxy values, determining the subject type, selecting the
186 appropriate vectorization function, and inputting these into the corresponding evaluation model to
187 predict relevant metrics. The base learners for the evaluation models can vary and include algorithms

such as linear models, Random Forests (RF), Multi-Layer Perceptrons (MLP), CatBoost, XGBoost, and LightGBM. Following the development of these evaluation models, an upper bound is calculated to check whether the models meet practical requirements and constraints.

# 5   Assessing assumptions

The IRIS assumption can be satisfied in data collection while the other assumptions may not be satisfied in real world. While the IIDE assumption could be trivially met by employing a random number generator as the outputs for IID outcome, this approach is devoid of practical utility, as it fails to reflect any substantive patterns or domain-specific knowledge required for realistic evaluation modeling. In order to assess the IIDE and unbiased assumptions, statistical analyses can be employed to verify necessary conditions under those assumptions, providing credibility to the theorems derived from this assumption for someone. It is important to note that these assessments do not conclusively prove the assumptions. Instead, they serve as methods to potentially refute the assumptions under specific conditions, rather than confirming its validity.

## 5.1   Assessing IIDE assumption

In highly controlled environments, such as CPU evaluation, the assumption can be rigorously satisfied by independently and randomly selecting tested chips and configurations from an identical distribution. However, in many open environments, the IIDE assumption is not directly testable on real-world data, such as individual medicine, and social experiment.

First, the normalized means of error subsets should follow a common Gaussian distribution, as implied by the central limit theorem. To test this, we randomly generate 30 subsets and apply the D'Agostino-Pearson test to assess whether their normalized means adhere to a single Gaussian distribution. Second, error distributions across different splits should be identical. We randomly partition the errors into 30 subsets and use the Kolmogorov-Smirnov test to determine whether the distributions across these subsets are statistically indistinguishable.

## 5.2   Assessing Unbiased assumption

Our effect error bounds requires unbiased evaluation models while sometime the evaluation model may not be unbiased. To assess the unbiasedness of the evaluation model, we employ the following statistical procedures: 1) Global Unbiasedness Assessment: We conduct a one-sample t-test on the entire dataset to determine if the model's errors have a mean of zero, indicating unbiasedness across the universal set; 2) Subset Unbiasedness Assessment: We randomly select 30 subsets from the dataset and perform one-sample t-tests on each to evaluate whether the mean errors within these subsets deviate significantly from zero. This step examines the consistency of the model's unbiasedness across different data partitions.

# 6   Experiment

We performed experiments for evaluation model without condition and evaluation model with condition respectively in 12 scenes of mini agent evaluation, including randomized experiment prediction, scientific simulation, advertising, insurance, and quantum trade. The detailed evaluation scenes (including experiment setting), assumption assessing, hyperparameter tuning, and evaluation results are listed in the appendix D, E, F, G.

## 6.1   Evaluation model

To demonstrate the generalizability of our evaluation model in agent space, we consider 11 distinct scenes. In these scenes, computational evaluation can be helpful to reduce experimental cost, shorten simulation time, or enhance the efficiency of advertising and insurance sales.

Although subjects are independently, randomly, and identically sampled from the subject space, they cannot be randomly deployed in real-world settings for direct measurement of benefit-related metrics (such as mortality, rehabilitation time) due to experimental limitation. Instead, our evaluation system defines the evaluation metric as the post-experiment metric on unseen test data with randomized

treatment. The metrics capture the causal effect from subject to post-experiment outcome, as established in Theorem 4. Empirical studies (Ai et al. [2022], Zhou et al. [2023], Zhou et al. [2024a], Cheng et al. [2022], Gentzel et al. [2021]) have also demonstrated the correlation between the agents' predictive performance on randomized experimental data and its performance following randomized deployment. Accordingly, we employ these predictive metrics to approximate randomized post-deployment performance, which can then be integrated into the AI decision via an appropriate utility function.

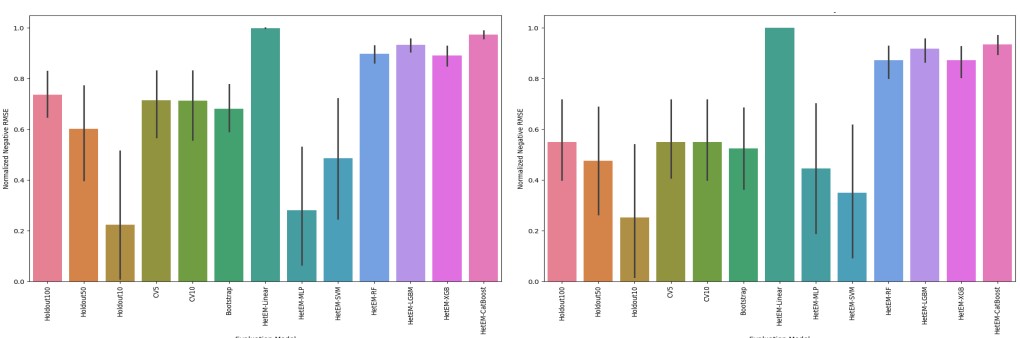

(a) Normalized negative RMSE when evaluation metric is ROC-AUC.

(b) Normalized negative RMSE when evaluation metric is ACC.

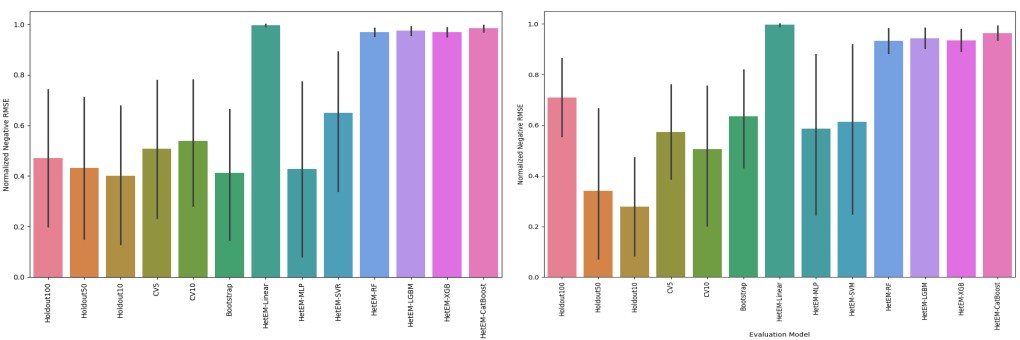

(c) Normalized negative RMSE when evaluation metric is RMSE.

(d) Normalized negative RMSE when evaluation metric is $R^2$.

Figure 3: Normalized empirical negative RMSE of evaluation models crossing scenes (6 scenes in figure 3a and figure 3b, 5 scenes in figure 3c and figure 3d. The confidence level of confidence interval bar is set as 95%. The baselines are holdout 100%, holdout 50%, holdout 10%, 5-fold cross-validation, 10-fold cross-validation, and bootstrap. We test different base learners (Linear, MLP, SVM/SVR, Random forest, XGBoost, LighGBM, and CatBoost) for heterogeneous subject space.

Across the 11 scenes, we compare our evaluation model against 6 existing validation methods that do not incorporate prior information from data. The normalized negative empirical RMSE is illustrated in Figure 3. Our best evaluation model (Het-Linear) achieves error reductions of 94.7% to 99.0% in RMSE, 81.7% to 95.5% in $R^2$, 24.1% to 77.0% in ROC-AUC, and 39.2% to 89.5% in ACC relative to the Holdout-100 validation method. This performance improvement is attributable to the effective modeling of the relationships among the subject, its proxy metrics, and the evaluation metrics. Notably, models that capture spurious correlations (MLP and SVR without hyperparameter tuning) fail to reduce evaluation error.

## 6.2 Conditional evaluation model

In order to demonstrate the generalizbility of conditional evaluation model on the condition space further, we introduce the trade backtesting task of A-share market in China to test the performance of conditional evaluation model. In this scene, the computational evaluation can be helpful to exclude the influence of hidden common cause between trade strategy and return of invest (RoI), and to reduce the evaluation cost of trade strategy's deployment.

The evaluation subjects are trade models whose inputs are last day's variables of the stock and output are one of three decisions (buy 1 hand, sell 1 hand, and hold) at open this day. The metric is Return of Invest of a subject in a time slot (10 days) in future. The condition is the stock's time-series variables in the last time slot. In this scene, we not only measure the subject and its metric, but also consider the pre-trade variables (condition) of the subject in an attempt to reduce evaluation error further. We use the unseen future data and performance of unseen subjects for conditional evaluation model testing.

Similarly, we build the evaluation systems by combining the real data and a float model whose input is subject vector and last day's variables of the stock, and output is a small float ratio of opening price. We want to use it to simulate the exclusion of hidden common cause on the stock market by the data without the randomized deployment of the subject.

| Backtesting method | RMSE of RoI |
|---|---|
| Baseline (Last10Days) | 3.067 |
| HetEM (Linear) | 2.163 |
| HetEM (CatBoost) | 0.527 |

Table 2: Performance of backtesting methods.

We compare our conditional evaluation models with a baseline backtesting methods as shown in the table 2. The reason we did not take the time cross-validation, k-fold cross-validation, and combinatorial symmetry cross-validation (Bailey et al. [2013]) into baselines is that those methods are train-valid fused rather than evaluation-targeted methods. When the validation part is separated, they will give the same trivial constant prediction which can not utilize the evaluation conditions for each time slot.

Our best conditional evaluation models (HetEM-CatBoost) for heterogeneous subjects reduced 89.4% evaluation errors comparing with the baseline method. Even the linear conditional evaluation model can reduce the 34.0% RMSE of estimated RoI. The average RMSE of estimated RoI in a future time slot is about 0.527, which may be reduced further when adding more factors of stocks, and designing novel neural network architectures. From the result of different evaluation methods, the computational evaluation approach has higher potential to solve the backtest over-fitting problem (Bailey et al. [2013]) in stock market.

## 6.3 Interpretability

In order to understand the contribution of subject vector, and proxy metric in the evaluation of our best evaluation model respectively, we visualize the Shapley value (Shapley [1997], Lundberg and Lee [2017]) for subject vector and proxy metric of all models as shown in figrue 4. The null set's value is set as the negative RMSE of holdout-100. From the visualization, the main contribution for heterogeneous subject whose metrics are ROC-AUC and ACC, is from the relationship between the proxy metric and true metric despite of an example in climate prediction scene. For heterogeneous subjects whose metrics are RMSE and $R^2$, the contribution of subject and proxy metric is almost equal to each other. It reveals the effectiveness of the introduced proxy module and subject vectorization module in our learning algorithm respectively.

In order to understand the contribution of the condition vector, subject vector, and proxy metrics in the evaluation of our best conditional evaluation model, we use Shapley values to explain the contribution of modules (Shapley [1997], Lundberg and Lee [2017]). The null set is set as the baseline backtesting method. The contribution rate of condition vector, subject vector, and proxy metrics for the HetEM(CatBoost) are 28.8% (0.729 uplift RMSE of RoI), 35.6% (0.905 uplift RMSE of RoI), and 35.6% (0.906 uplift RMSE of RoI) respectively. It reveals the importance of evaluation condition to reduce the evaluation errors in the A-share trade scenes of China.

## 6.4 Evaluation cost

The primary advantage of our computational evaluation method is its efficiency and low cost. Rather than conducting an unbounded number of experiments for infinitely many subjects (e.g., AI models or other agents), our approach only requires the collection of initial data, training an evaluation model, and subsequently applying it in real-world scenes.

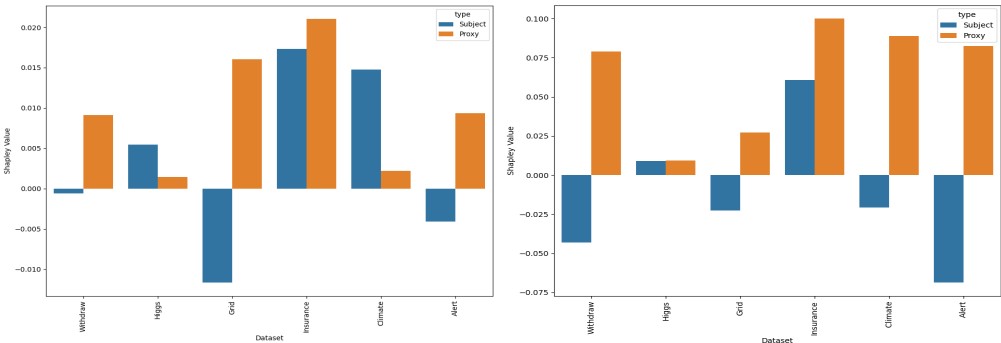

(a) Shapley Value of Subject and Proxy when metric is ROC-AUC.

(b) Shapley Value of Subject and Proxy when metric is ACC.

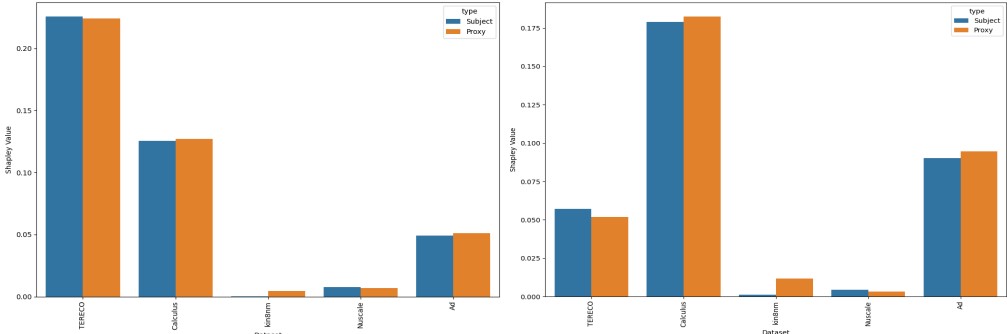

(c) Shapley Value of Subject and Proxy when metric is RMSE.

(d) Shapley Value of Subject and Proxy when metric is $R^2$.

Figure 4: Shapley value of subject vector and proxy metrics on 11 scenes (6 scenes in figure 4a and figure 4b, 5 scenes in figure 4c and figure 4d. The outcome of null set is set as the performance of holdout-100, and other outcomes are the negative RMSE of the linear evaluation models.

The acceleration ratio of time achieved by our computational method ranges from 1,000 to 10,000,000 per evaluation subject in the 12 scenes as shown in appendix H. Although experimental and simulation-based evaluations can be accelerated via parallel processing (excluding additional evaluation costs), the same parallelization strategies can be readily applied to computational evaluation approach.

## 7 Limitation and conclusion

**Limitation.** First, our theoretical guarantees rely on three assumptions, which may not hold for all real-world scenes and agents. Second, while our method has been validated across multiple domains, its generalization in extremely high-dimensional settings (such as LLM) require further investigation. Third, lower error and tighter bound are required to apply evaluation models into real applications with small sample. Four, scalability of evaluation models across scenes is another problem in future study. Five, the post-experiment metric (such as mortality) of *randomized deployed mini-agents* with high utility should be collected to increase the value of computational evaluation.

**Conclusion.** In this work, we propose a novel computational framework for the evaluation of mini-agents that rigorously derives upper bounds on generalized evaluation error and effect error, and employs a meta-learning strategy to address heterogeneity in agent space. Our extensive experimental results, conducted over 12 diverse scenes—including individual medicine, scientific simulation, social experiments, business activity, and quantum trade—demonstrate significant error reductions (ranging from 24.1% to 99.0%) and substantial acceleration (up to $10^7$) compared to traditional evaluation methods. These findings highlight the potential of our technical route to drastically reduce evaluation costs for rapid model iteration. Future work will focus on more benefit-related metrics, multi-source inputs, novel neural architectures for evaluation tasks and enhancing the validation of key assumption. The upper bound table and paradigm of computational evaluation were given in the appendix I and J.

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

# A  Investigation of individual medical AI in randomized controlled trial

From January 2022 and January 2023, there are total 380 completed phase III interventional trials (adult) with results and study documents on ClinicalTrials.gov. The statistical estimates can be categorized into six classes: geometric mean, mean, square mean, median, rank, and percentage, as shown in the figure 5. It is important to note that these estimates can not reflect the probability of a group or an individual benefiting from the treatment. For example, for treatment effect with Gaussian distribution, the mean difference can still be significant even the treatment is harmful for $49\%$ and benefited for $51\%$ if $n$ is large enough. The randomized controlled trails can not be applied to development of individual treatment AI directly.

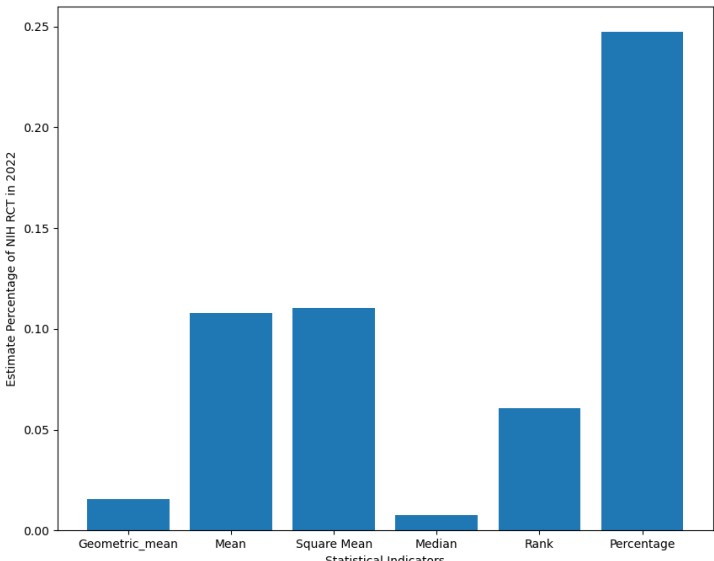

Figure 5: Estimates percentage of completed phase III interventional trials (adult) with results and study documents on https://clinicaltrials.gov.

Another query was conducted on ClinicalTrials.gov using the keyword **"Artificial Intelligence"** under the *Condition or Disease* field. Filters were applied to select only **completed**, **interventional studies**, **with posted results**, and **available statistical analysis plans**, with **no restriction on completion date** at January of 2025.

Only a total of 13 studies were retrieved. After manually excluding *single-arm trials* (i.e., studies without control groups or randomized design), 7 randomized clinical trials (RCTs) (Nayak et al. [2023], Abramoff et al. [2023], Lv et al. [2023], MacNeill et al. [2024], Eng et al. [2021], Piette et al. [2022], Mohr et al. [2019]) remained that investigated the clinical efficacy of AI-based healthcare interventions.

Among these, only 3 trials reported statistically significant primary outcomes, including:

- An AI-powered blood glucose reminder system (Nayak et al. [2023]),

- The Wysa AI chatbot for mental health support (MacNeill et al. [2024]),

- An AI system for bone age prediction (Eng et al. [2021]).

These results highlight the *challenge of translating AI agent systems into demonstrably effective clinical interventions*, emphasizing the need for rigorous evaluation, robust study design, and careful integration into complex healthcare workflows.

# B Investigation of agent evaluation

## B.1 Existed agent evaluation works

The theoretical foundations of agent evaluation, particularly with causal guarantees, remain underdeveloped. To date, we are not aware of any systematic theory that rigorously supports causal inference for agent performance evaluation. While several benchmarking efforts exist, they primarily focus on performance measurement rather than causal understanding. MLAgentBench (Huang et al. [2024]) uses 13 tasks (including Canonical Tasks, Classic Kaggle, Kaggle Challenges, Recent Research, and Code Improvement) to evaluate the performance of large language model. SUPER Benchmark (Bogin et al. [2024]) is used to evaluate the agents' ability to reproduce results from research repositories. AI Clinician (Komorowski et al. [2018]) and AI Hospital (Fan et al. [2025]) use synthetic data to test the performance of clinical agents. Jia et al. [2024] explores the LLMs' output decisions' alignment with human norms and ethical expectations. However, a common limitation across these works is the lack of control over unobserved confounders, such as participants' pre-experiment training or researchers' design biases. These hidden common causes may simultaneously influence both the deployed agent and its measured performance in real-world settings. As a result, these benchmarks cannot identify the (conditional) causal effect of an agent on real-world outcomes—such as clinical mortality—limiting their utility for causal evaluation.

## B.2 Causal mini agent evaluation and potential outcome

Despite targeting different problem settings, the causal computational evaluation framework and the potential outcomes framework (Imbens and Rubin [2015]) for agent evaluation share one key connection and exhibit two fundamental differences. The connection lies in the fact that the unconfoundedness (ignorability/exchangeability) assumption in potential outcomes can be directly implied by the IRIS assumption. Furthermore, IRIS can be satisfied in practice by deploying mini agents in a randomized manner. The first major difference is that the Stable Unit Treatment Value Assumption (no interference and consistency)—a common requirement in potential outcome formulations—is not necessary under our framework, as long as the IIDE (Independent and Identically Distributed Evaluations) assumption holds. The second key difference is that the positivity assumption, which typically requires a non-zero probability of receiving each treatment, is also not required. Nevertheless, the causal effect of mini agents on evaluation metrics can still be bounded in the limit, as formally established in Theorem 4. These distinctions suggest that the proposed framework can be viewed as a generalized and more flexible extension of the potential outcomes framework, tailored specifically for the causal evaluation of mini agent behaviors.

## B.3 Existed causal inference benchmark and packages

Although a number of benchmark datasets and open-source packages have been developed to address general causal inference problems—such as CausalML (Zhao and Liu [2023]), EconML (Battocchi et al. [2019]), DoWhy (Blöbaum et al. [2024], Sharma and Kiciman [2020]), CauseBox (paras2612), CausalNex (quantumblacklabs), Causal Curve (Kobrosly [2020]), CausalDiscovery (Kalainathan et al. [2020]), pcalg (Kalisch et al. [2012]), bnlearn (Scutari [2010]), TETRAD (Ramsey et al.), CausalityBenchmark (Shimoni et al. [2018]), JustCause (inovex), CausalEffect (Tikka and Karvanen [2017]), Ananke (Lee et al. [2023]), Dagitty (Textor et al. [2011]), YLearn (Bochen Lyu [2022]), CausalImpact (Google), causal-learn (Zheng et al. [2024]), gCastle (Zhang et al. [2021]), dosearch (Tikka et al. [2021]), CEE (Li), awesome-causality-identification (YAN [2025]) —these tools are primarily designed for causal discovery or treatment effect estimation. Despite extensive investigation, we did not identify any existing algorithms or frameworks that directly address the problem of causal evaluation of agents, particularly in the context of evaluating agent-driven decision-making processes.

# C  Proofs

**Theorem 1.** *Upper bound. Given any evaluation model $\hat{f}$, $P(E_{gen}(\hat{f}) < E_{emp}(\hat{f}) + \sqrt{\frac{1}{2n}\ln(\frac{1}{\sigma})}) \geq 1 - \sigma$ where $n$ is number of independent identical distributed error (IIDE) measurements, $0 < 1 - \sigma < 1$ is confidence.*

*Proof.* Denote $\epsilon_0 \cong \sqrt{\frac{1}{2n}\ln(\frac{1}{\sigma})}$, inequality in theorem 1 is equivalent to

$$P(E_{gen}(\hat{f}) - E_{emp}(\hat{f}) < \epsilon_0) \geq 1 - \sigma$$

. Because event $E_{gen}(\hat{f}) - E_{emp}(\hat{f}) < \epsilon_0$ and event $E_{gen}(\hat{f}) - E_{emp}(\hat{f}) \geq \epsilon_0$ are complementary events, so

$$P(E_{gen}(\hat{f}) - E_{emp}(\hat{f}) < \epsilon_0) = 1 - P(E_{gen}(\hat{f}) - E_{emp}(\hat{f}) \geq \epsilon_0)$$

. Bring it in, we have

$$1 - P(E_{gen}(\hat{f}) - E_{emp}(\hat{f}) \geq \epsilon_0) \geq 1 - \sigma$$

. Then it can be simplified as

$$P(E_{gen}(\hat{f}) - E_{emp}(\hat{f}) \geq \epsilon_0) \leq \sigma$$

. Because evaluation error is distributed on $[0,1]$, and Hoeffding inequality (Hoeffding [1963]),

$$P(E_{gen}(\hat{f}) - E_{emp}(\hat{f}) \geq \epsilon_0)$$

$$\leq 2e^{-\frac{2\epsilon_0^2 n^2}{\Sigma_{i=1}^n (1-0)^2}} = 2e^{-\frac{2(\sqrt{\frac{1}{2n}(\ln(\frac{1}{\sigma}))})^2 n^2}{\Sigma_{i=1}^n (1-0)^2}} = \sigma$$

$\square$

**Theorem 2.** *Strict causal advantage. Given $\{c_i, s_i, m_i\}_{i=1}^n$, where $c$ is evaluation condition, $s$ is independent from $c$, and $e$, and $s$ is independently, randomly, identically sampled (IRIS) from distribution $Pr(S)$, $m$ is generate by an unknown and inaccessible function $m = f(c, e, s)$, $e$ is other unlisted random variable, evaluation error function $L$ is mean square error, $\forall \hat{f}_1, \forall \hat{f}_2$ where $\hat{f}_1$ and $\hat{f}_2$ are unbiased estimation of $f$ given $c$ (CU), we have **Efficiency**: if $E_{gen}(\hat{f}_1) < E_{gen}(\hat{f}_2)$ for any $s$, then $\forall s_a \in \mathbf{S}$, $\forall s_b \in \mathbf{S}$, $\forall c \in \mathbf{C}$, $E_{gen}(\hat{f}_1(c, s_a) - \hat{f}_1(c, s_b)) < E_{gen}(\hat{f}_2(c, s_a) - \hat{f}_2(c, s_b))$; **Consistency**: if $\lim_{\hat{f} \to f} E_{gen}(\hat{f}) = 0$, then $\forall s_a \in \mathbf{S}, \forall s_b \in \mathbf{S}, \forall c \in \mathbf{C}, \lim_{\hat{f} \to f} E_{gen}(\hat{f}(c, s_a) - \hat{f}(c, s_b)) = 0$.*

*Proof.* $\forall k = 1, 2$, denote $\epsilon_k(c, s, e) = \hat{f}_k(c, s) - f(c, e, s)$, according to conditional unbiasness, we have $E(\epsilon_k(c, s)) = 0$. The error is denoted as

$$\Delta \hat{f}_k(s) = \hat{f}_k(c, s_a) - \hat{f}_k(c, s_b) - (f(c, s_a, e) - f(c, s_b, e)) = \epsilon_k(c, s_a, e) - \epsilon_k(c, s_b, e)$$

, the square error is denoted as

$$(\Delta \hat{f}_k)^2 = (\epsilon_k(c, s_a, e) - \epsilon_k(c, s_b, e))^2$$

, the mean square error is denoted as

$$E((\Delta \hat{f}_k)^2) = E(\epsilon_k(c, s_a, e)^2) + E(\epsilon_k(c, s_b, e)^2) - 2E(\epsilon_k(c, s_a, e)\epsilon_k(c, s_b, e))$$

. According to the IRIS assumption and condition unbiasness assumption, the third term

$$E(\epsilon_k(c, s_a)\epsilon_k(c, s_b)) = E_c(E(\epsilon_k(c, s_a, e)\epsilon_k(c, s_b, e)|c))$$

$$= E_c(E(\epsilon(c, s_a)|c)E(\epsilon(c, s_b)|c)) = E_c(0 * 0) = E_c(0) = 0$$

, then because $\forall s$, $E_{gen}(\hat{f}_1) < E_{gen}(\hat{f}_2)$, that is $E(\epsilon_1(c, s, e)^2) < E(\epsilon_2(c, s, e)^2)$ for any $s$, then

$$E((\Delta \hat{f}_1)^2) = E(\epsilon_1(c, s_a, e)^2) + E(\epsilon_1(c, s_b, e)^2)$$

$$< E(\epsilon_2(c, s_a, e)^2) + E(\epsilon_2(c, s_b, e)^2) = E((\Delta \hat{f}_2)^2)$$

, which can be write as $E_{gen}(\hat{f}_1(c, s_a) - \hat{f}_1(c, s_b)) < E_{gen}(\hat{f}_2(c, s_a) - \hat{f}_2(c, s_b))$ when error function is mean square error.

Following, the consistency can be proved spontaneously by the Squeeze Theorem in Calculus.  $\square$

**Theorem 3.** *Causal bound with positivity.* $\forall s_a \in S, \forall s_b \in S$, given any unbiased evaluation model $\hat{f}(c, s_a)$ and $\hat{f}(c, s_b)$,

$$P(E_{gen}(\Delta \hat{f}) < 2 \max\{E_{emp}(a) + \sqrt{\frac{1}{2n_a} \ln(\frac{2}{\sigma})}), E_{emp}(b) + \sqrt{\frac{1}{2n_b} \ln(\frac{2}{\sigma})}\}) \geq 1 - \sigma$$

*, where $\Delta \hat{f} = \hat{f}(c, s_a) - \hat{f}(c, s_b)$, $n_a$ and $n_b$ are number of independently, randomly, identically sampled error measurements (IIDE) where $s = s_a$ and $s = s_b$, $0 < 1 - \sigma < 1$ is confidence, evaluation error function L is mean square error ranged from 0 to 1.*

*Proof.* Because evaluation error function is mean square error,

$$E_{gen}(\Delta \hat{f}) \cong E((\Delta \hat{f} - \Delta f)^2) = E((\hat{f}(c, s_a) - \hat{f}(c, s_b) - (f(c, s_a, e) - f(c, s_b, e)))^2)$$

. Denote $\epsilon_a(c, e) \cong \hat{f}(c, s_a) - f(c, s_a, e)$ and $\epsilon_b(c, e) \cong \hat{f}(c, s_b) - f(c, s_b, e)$,

$$E_{gen}(\Delta \hat{f}) = E(\epsilon_a(c, e)^2 + \epsilon_b(c, e)^2 - 2\epsilon_a(c, e)\epsilon_b(c, e))$$

. Due to properties of expectation,

$$E_{gen}(\Delta \hat{f}) = E(\epsilon_a(c, e)^2) + E(\epsilon_b(c, e)^2) - 2E(\epsilon_a(c, e)\epsilon_b(c, e))$$

. Due to IIDE assumption, the sub population of errors where $s = a$ and $s = b$ is also independently sampled,

$$E_{gen}(\Delta \hat{f}) = E(\epsilon_a(c, e)^2) + E(\epsilon_b(c, e)^2) - 2E(\epsilon_a(c, e))E(\epsilon_b(c, e))$$

. Due to the unbiased assumption,

$$E_{gen}(\Delta \hat{f}) = E(\epsilon_a(c, e)^2) + E(\epsilon_b(c, e)^2)$$

. Denote $\epsilon_0 = 2 \max\{\nu_a, \nu_b\}$ where $\nu_a = E_{emp}(a) + \sqrt{\frac{1}{2n_a} \ln(\frac{2}{\sigma})}$ and $\nu_b = E_{emp}(b) + \sqrt{\frac{1}{2n_b} \ln(\frac{2}{\sigma})}$, due to Bonferroni inequality (Galambos [1977]),

$$P(E_{gen}(\Delta \hat{f}) < \epsilon_0) = P(E(\epsilon_a(c, e)^2) + E(\epsilon_b(c, e)^2) < \epsilon_0)$$

$$\geq P(E(\epsilon_a(c, e)^2) < \frac{\epsilon_0}{2} \bigcap E(\epsilon_b(c, e)^2) < \frac{\epsilon_0}{2})$$

$$\geq P(E(\epsilon_a(c, e)^2) < \frac{\epsilon_0}{2}) + P(E(\epsilon_b(c, e)^2) < \frac{\epsilon_0}{2}) - 1$$

$$\geq P(E(\epsilon_a(c, e)^2) < \nu_a) + P(E(\epsilon_b(c, e)^2) < \nu_b) - 1$$

$$\geq 1 - \frac{\sigma}{2} + 1 - \frac{\sigma}{2} - 1$$

$$= 1 - \sigma$$

$\square$

**Theorem 4.** *Causal bound with non-positivity.* *Given unbiased evaluation model $\hat{f}$, then $\forall s_a \in S, \forall s_b \in S$,*

$$P(E_{gen}(\Delta \hat{f}) < 2(E_{emp}(\hat{f}) + \sqrt{\frac{1}{2n} \ln(\frac{1}{\sigma})})) \geq 1 - \sigma$$

*, where $\Delta \hat{f} = \hat{f}(c, s_a) - \hat{f}(c, s_b)$, $n$ is number of independently, randomly, identically sampled error measurements (IIDE), $0 < 1 - \sigma < 1$ is confidence, s is independently, randomly, identically sampled (IRIS), evaluation error function L is mean square error ranged from 0 to 1.*

*Proof.* According to same proof and denotation in theorem 3,

$$E_{gen}(\Delta\hat{f}) = E(\epsilon_a(c,e)^2) + E(\epsilon_b(c,e)^2) - 2E(\epsilon_a(c,e))E(\epsilon_b(c,e))$$

. Due to IIDE and IRIS assumption, the sub population of errors where $s = a$ and $s = b$ is also independently sampled (IIDE) identically following the whole population (IRIS),

$$E_{gen}(\Delta\hat{f}) = 2E(\epsilon(c,e)^2) - 2E(\epsilon(c,e))^2$$

. Due to the unbiased assumption,

$$E_{gen}(\Delta\hat{f}) = 2E(\epsilon(c,e)^2)$$

. According to theorem 1, let evaluation error function be MSE,

$$P(E(\epsilon(c,e)^2) \leq \sum_{i=1}^{n}(\hat{f}(c_i,s_i) - m_i)^2 + \sqrt{\frac{1}{2n}\ln(\frac{1}{\sigma})}) \geq 1 - \sigma$$

where $m_i$ is true metric. Due to the property of equivalent events,

$$P(2E(\epsilon(c,e)^2) \leq 2(\sum_{i=1}^{n}(\hat{f}(c_i,s_i) - m_i)^2 + \sqrt{\frac{1}{2n}\ln(\frac{1}{\sigma})})) \geq 1 - \sigma$$

, which can be rewrite as

$$P(E_{gen}(\Delta\hat{f}) \leq 2(E_{emp}(\hat{f}) + \sqrt{\frac{1}{2n}\ln(\frac{1}{\sigma})})) \geq 1 - \sigma$$

. $\square$

# D  Evaluation scenes

We introduce 12 scenes to test the performance of our (conditional) evaluation model: Alert (Wilson et al. [2021]), Withdraw (Wilson et al. [2023]), Grid-S (Schäfer et al. [2016]), Higgs-S (Baldi et al. [2014]), Insurance (Kumar [2023]), Climate-S (Lucas et al. [2013]), TERECO (Li et al. [2022]), Calculus (Kramer et al. [2023]), Ad (Gokagglers [2018]), kin8nm-S (Fischer [2022]), NuScale-S (Huu Tiep [2024]), and Quantum trade. The evaluation subject (agent) and evaluation metric in those applied scenes is listed in the table 4. All the data are collected from real experiments or simulation. As mentioned in the section 6, we use the prediction performance in randomized experiment to approximate the post-experiment performance of randomized deployed mini agents. We also listed the website and license of the mentioned asset in the following table 3.

| Scenes | Website | License |
|---|---|---|
| Alert | https://datadryad.org/dataset/doi:10.5061/dryad.4f4qrfj95 | CC0 1.0 |
| Withdraw | https://datadryad.org/dataset/doi:10.5061/dryad.kh189327p | CC0 1.0 |
| Grid-S | https://www.openml.org/search?type=data&status=active&id=44973 | CC BY 4.0 |
| Higgs | https://www.openml.org/search?type=data&status=active&id=42769 | Public |
| Insurance | https://www.kaggle.com/datasets/arun0309/customer | Unknown |
| Climate | https://www.openml.org/search?type=data&status=active&id=40994 | Public |
| TERECO | https://datadryad.org/dataset/doi:10.5061/dryad.59zw3r27n | CC0 1.0 |
| Calculus | https://datadryad.org/dataset/doi:10.5061/dryad.kkwh70s95 | CC0 1.0 |
| Ad | https://www.kaggle.com/datasets/loveall/clicks-conversion-tracking | Other |
| kin8nm | https://www.openml.org/search?type=data&status=active&id=44980 | Public |
| NuScale | https://archive.ics.uci.edu/dataset/1091 | CC BY 4.0 |

Table 3: Website and license of the mentioned asset.

The heterogeneous mini agent space is configured as a combination of a linear mini agent space (logistic regression or linear regression) and a non-linear mini agent space (MLP Classifier or MLP Regressor). The agent's input dimension is the number of features, and output dimension is the number of the targets. The MLP's hidden layer size limit is set as 8.

| Scene | Evaluation Subject (Agent) | Application | Metric |
|---|---|---|---|
| Individual Treatment | input is patients' record, and random pop-up, output is mortality | intelligent Alert, and Withdrawal for AKI | ROC-AUC ACC |
| | input is patients' record, and random therapy, output is walk distance | Covid-19 recovery (TERECO) | RMSE $R^2$ |
| Scientific Simulation | input is setting of fission reactor, output is k-inf and PPPF | nuclear power plant (Nuscale) | RMSE $R^2$ |
| | input is particle features output is whether it is higgs | higgs detection (Higgs) | ROC-AUC ACC |
| | input is initial parameter of climate, output is whether the climate crash | climate simulation (Climate) | ROC-AUC ACC |
| | input is joint moving of robotic arm, output is distance to target | robot action (kin8nm) | RMSE $R^2$ |
| | input is parameter of electric grid, output is whether the grid crash | grid simulation (Grid) | ROC-AUC ACC |
| Social Experiment | input is students' record, and random class output is students' grading | individual teaching (Calculus) | RMSE $R^2$ |
| Business | input is customers' record, and random recommendation output is buy-in decision of custom | sell insurance (Insurance) | ROC-AUC ACC |
| | input is feature of a group, and random recommendation output is buy-in decision of custom | advertisement (Ad) | RMSE $R^2$ |
| Quantum Trade | input is stock's feature last day, output is decision for the stock | A-share trade | RoI |

Table 4: Detailed setting of scene.

The IRIS sampling of the agent is two steps. The first step is the agent type sampling with 0.5 probability to choose linear model and 0.5 probability to choose MLP model. The second step is to choose a specific model in the sub space. For both linear agent and MLP agent, we use Normal distribution $N(0, 1)$ to determine the parameter of the models. The outcome when we calculate Shapley value is set as the negative evaluation errors.

The samples number of scenes Alert, Withdrawal, Higgs, Climate, Grid, Insurance, TERECO, Nuscale, kin8nm, Calculus, Ad, and A-share trade are listed in table 5. The 7 kinds of base learners we used is Linear, MLP, SVM/SVR, RF (Breiman [2001]), LGBM (Aziz et al. [2022]), XGBoost (Chen and Guestrin [2016]), and CatBoost (Prokhorenkova et al. [2018]). All the computation is on the one 13-inch MacBook Pro 2020 with Apple M1 chip and 16GB memory in 2 days.

| Scenes | Train Samples | Test Samples |
|---|---|---|
| Alert | $1600 * 5911 * 0.2 = 1891520$ | $400 * 5911 * 0.2 = 472880$ |
| Withdrawal | $1600 * 4998 * 0.2 = 1599360$ | $400 * 4998 * 0.2 = 399840$ |
| Higgs | $1600 * 2000 * 0.2 = 640000$ | $400 * 2000 * 0.2 = 160000$ |
| Climate | $1600 * 540 * 0.2 = 172800$ | $400 * 540 * 0.2 = 43200$ |
| Grid | $1600 * 10000 * 0.2 = 3200000$ | $400 * 10000 * 0.2 = 800000$ |
| Insurance | $1600 * 45211 * 0.2 = 14467520$ | $400 * 45211 * 0.2 = 3616880$ |
| TERECO | $1600 * 104 * 0.2 = 33280$ | $400 * 104 * 0.2 = 8320$ |
| Nuscale | $1600 * 360 = 576000$ | $400 * 360 = 144000$ |
| kin8nm | $1600 * 8192 * 0.2 = 2621440$ | $400 * 8192 * 0.2 = 655360$ |
| Calculus | $1600 * 672 * 0.2 = 215040$ | $400 * 672 * 0.2 = 53760$ |
| Ad | $1600 * 1143 * 0.2 = 365760$ | $400 * 1143 * 0.2 = 91440$ |
| Trade | $160 * 200 * 30 = 960000$ | $40 * 200 * 20 = 160000$ |

Table 5: Evaluation sample numbers in different scenes.

### D.1 Scenes to test evaluation model

For 11 scenes to test evaluation model, 2000 agents was randomly sampling from heterogeneous agent space. 20% data was used to build the real systems to generate true evaluation metrics and other 80% data was used to get the proxy metrics of a agent. When the evaluation samples are collected from the built real systems, 20% samples are used for evaluation model testing, and other 80% samples are for evaluation model learning. The experiment is performed 30 times. All the null value was removed from the raw data in preprocessing. The classical features are encoded by the label encoder to reduce the computation. The text classical features are encoded by one-hot encoder.

The proxy metrics we used for Alert, Withdrawal, Higgs, Climate, Grid, Insurance are ROC-AUC, accuracy, recall, precision, $F_1$ score, and PR-AUC. The proxy metrics we used for TERECO, Nuscale, kin8nm, Calculus, Ad are RMSE, $R^2$, MAE, MAPE, and MSE.

### D.2 Scenes to test conditional evaluation model

We collected the A-share data of China from 2024-8 to 2024-11 (89 days) by an open source tool Ak-Share. The collected day-wise features of stocks are "Profit Ratio", "Average Cost", "90 Cost Low", "90 Cost High", "90 Episode Medium", "70 Cost Low", "70 Cost High", "70 Episode Medium", "Closing", "Opening", "Highest", "Lowest", "Transaction Volume", "Transaction Amount", "Amplitude". The strategy is to buy 1 hand, sell 1 hand, or hold by the agent following the open price of the market. The slippage and commission is not considered in this work. If the stock was limit up (limit down), then we can only hold even the model decision is to buy (sell). The agent's input is the feature of the stock last day and output is the decision this day. The agent will be deployed and run for 1 time slot (10 trading days, about 2 weeks), and the evaluation condition is feature of the stock in the last time slot. At the end of deployment in a time slot, we will sell all the hold stocks. When the end of the slot is limit down at open, we assume that it will be sold by 90 percents of open price in future.

In order to simulate the direct effect from the agent itself to the sensitive market, we add a float model which input is the vectorized agents and output is float rate of open price ranged from -0.1% to +0.1%.

We use data of 200 stocks with minimal ID from all 3214 stocks due to our memory limit. The start of train day is from day 10 to day 41, the start of test day is from 41 to 61. The sampling number of agent is 200. The experiment was performed 5 times to reduce the influence of randomness. The proxy metric is the last 10 days' RoI.

## E Assumption assessment

Due to the limitation of computation resources and the difference of addressed problems, we do not use the deep learning models for observation data as base learners of our evaluation models, such as CEVAE (Louizos et al. [2017]), BNR (Li and Fu [2017]), BNN (Johansson et al. [2016]), SITE (Yao et al. [2018]), GANITE (Yoon et al. [2018]), DeepMatch (Kallus [2020]), Dragonnet (Shi et al. [2019]), TARNet/CFR (Shalit et al. [2017]), TabPFN (Hollmann et al. [2025]), deconfounder (Wang and and [2019]), time-series deconfounder (Bica et al. [2020]). Table 6 and table 7 are assumption assessment results of 7 kinds of evaluation models with swift base learners for continuous and discrete target respectively. From the tables, we can see that the IIDE assumption (IID check and ID check) and unbiased evaluation model assumption (Bias Check and GroupBias Check) was not violated obviously with a high probability (>0.8) except the cases that using MLP and SVR as base regression learners of HetEM.

| Ratio $p >= 0.05$ | TERECO | Calculus | kin8nm | nuscale | ad |
|---|---|---|---|---|---|
| Het(Linear)-IID-RMSE | 0.93 | 0.87 | 0.97 | 0.9 | 0.97 |
| Het(Linear)-IID-$R^2$ | 0.93 | 1.0 | 0.9 | 0.93 | 0.83 |
| Het(Linear)-ID-RMSE | 1.0 | 0.47 | 0.87 | 1.0 | 0.93 |
| Het(Linear)-ID-$R^2$ | 0.97 | 0.43 | 0.93 | 0.97 | 0.87 |
| Het(Linear)-Bias-RMSE | 0.9 | 0.43 | 0.77 | 0.8 | 0.77 |
| Het(Linear)-Bias-$R^2$ | 0.9 | 0.33 | 0.8 | 0.77 | 0.77 |
| Het(Linear)-GroupBias-RMSE | 0.92 | 0.85 | 0.87 | 0.87 | 0.85 |

| | | | | | |
|---|---|---|---|---|---|
| Het(Linear)-GroupBias-$R^2$ | 0.90 | 0.85 | 0.86 | 0.84 | 0.89 |
| Het(MLP)-IID-RMSE | 1.0 | 0.93 | 0.83 | 1.0 | 0.9 |
| Het(MLP)-IID-$R^2$ | 0.9 | 0.97 | 0.93 | 0.97 | 0.97 |
| Het(MLP)-ID-RMSE | 0.73 | 0.37 | 0.87 | 0.73 | 0.77 |
| Het(MLP)-ID-$R^2$ | 0.9 | 0.3 | 0.77 | 0.77 | 0.67 |
| Het(MLP)-Bias-RMSE | 0.5 | 0.26 | 0.57 | 0.57 | 0.5 |
| Het(MLP)-Bias-$R^2$ | 0.57 | 0.17 | 0.53 | 0.57 | 0.57 |
| Het(MLP)-GroupBias-RMSE | 0.68 | 0.40 | 0.86 | 0.70 | 0.72 |
| Het(MLP)-GroupBias-$R^2$ | 0.70 | 0.31 | 0.71 | 0.67 | 0.68 |
| Het(SVR)-IID-RMSE | 0.97 | 0.97 | 0.97 | 1.0 | 0.87 |
| Het(SVR)-IID-$R^2$ | 0.93 | 0.9 | 1.0 | 0.9 | 0.97 |
| Het(SVR)-ID-RMSE | 0.0 | 0.0 | 0.27 | 0.4 | 0.1 |
| Het(SVR)-ID-$R^2$ | 0.0 | 0.0 | 0.0 | 0.2 | 0.0 |
| Het(SVR)-Bias-RMSE | 0.0 | 0.0 | 0.1 | 0.17 | 0.03 |
| Het(SVR)-Bias-$R^2$ | 0.0 | 0.23 | 0.0 | 0.03 | 0.0 |
| Het(SVR)-GroupBias-RMSE | 0.0 | 0.01 | 0.19 | 0.34 | 0.11 |
| Het(SVR)-GroupBias-$R^2$ | 0.0 | 0.44 | 0.00 | 0.19 | 0.02 |
| Het(RF)-IID-RMSE | 0.97 | 0.93 | 0.97 | 0.97 | 0.93 |
| Het(RF)-IID-$R^2$ | 1.0 | 0.97 | 0.97 | 1.0 | 0.97 |
| Het(RF)-ID-RMSE | 0.93 | 0.87 | 0.9 | 0.97 | 0.83 |
| Het(RF)-ID-$R^2$ | 0.93 | 0.93 | 0.93 | 0.9 | 0.73 |
| Het(RF)-Bias-RMSE | 0.93 | 0.57 | 0.9 | 0.9 | 0.8 |
| Het(RF)-Bias-$R^2$ | 0.83 | 0.63 | 0.93 | 0.87 | 0.8 |
| Het(RF)-GroupBias-RMSE | 0.87 | 0.85 | 0.91 | 0.90 | 0.86 |
| Het(RF)-GroupBias-$R^2$ | 0.89 | 0.86 | 0.91 | 0.88 | 0.86 |
| Het(LGBM)-IID-RMSE | 0.87 | 0.93 | 0.97 | 1.0 | 0.93 |
| Het(LGBM)-IID-$R^2$ | 1.0 | 0.97 | 0.97 | 0.93 | 0.97 |
| Het(LGBM)-ID-RMSE | 0.93 | 0.9 | 0.9 | 0.97 | 0.83 |
| Het(LGBM)-ID-$R^2$ | 0.9 | 0.93 | 0.83 | 0.97 | 0.93 |
| Het(LGBM)-Bias-RMSE | 0.9 | 0.7 | 0.8 | 0.9 | 0.77 |
| Het(LGBM)-Bias-$R^2$ | 0.93 | 0.7 | 0.77 | 0.9 | 0.83 |
| Het(LGBM)-GroupBias-RMSE | 0.89 | 0.85 | 0.90 | 0.88 | 0.85 |
| Het(LGBM)-GroupBias-$R^2$ | 0.89 | 0.86 | 0.97 | 0.88 | 0.87 |
| Het(XGBoost)-IID-RMSE | 1.0 | 0.93 | 0.93 | 1.0 | 0.87 |
| Het(XGBoost)-IID-$R^2$ | 0.93 | 0.93 | 0.93 | 0.97 | 0.93 |
| Het(XGBoost)-ID-RMSE | 0.93 | 0.9 | 1.0 | 0.93 | 0.97 |
| Het(XGBoost)-ID-$R^2$ | 0.93 | 0.9 | 0.93 | 0.93 | 0.87 |
| Het(XGBoost)-Bias-RMSE | 0.97 | 0.53 | 0.77 | 0.87 | 0.87 |
| Het(XGBoost)-Bias-$R^2$ | 0.83 | 0.5 | 0.9 | 0.87 | 0.83 |
| Het(XGBoost)-GroupBias-RMSE | 0.92 | 0.84 | 0.83 | 0.86 | 0.88 |
| Het(XGBoost)-GroupBias-$R^2$ | 0.87 | 0.82 | 0.90 | 0.84 | 0.88 |
| Het(CatBoost)-IID-RMSE | 0.87 | 0.97 | 0.97 | 1.0 | 1.0 |
| Het(CatBoost)-IID-$R^2$ | 0.87 | 0.97 | 0.93 | 0.93 | 1.0 |
| Het(CatBoost)-ID-RMSE | 0.93 | 0.87 | 0.93 | 0.97 | 0.97 |
| Het(CatBoost)-ID-$R^2$ | 0.97 | 0.9 | 0.93 | 0.97 | 1.0 |
| Het(CatBoost)-Bias-RMSE | 0.93 | 0.6 | 0.97 | 0.8 | 0.83 |
| Het(CatBoost)-Bias-$R^2$ | 0.93 | 0.63 | 0.93 | 0.8 | 0.87 |
| Het(CatBoost)-GroupBias-RMSE | 0.90 | 0.90 | 0.92 | 0.87 | 0.87 |
| Het(CatBoost)-GroupBias-$R^2$ | 0.90 | 0.87 | 0.90 | 0.86 | 0.88 |

Table 6: The ratio that the p-value of test is larger than or equal to 0.05 in the 30 repeated experiments for regression base learners. The name of each row is "EvaluationModelName"-"TestName"-"MetricName". The name of each column is the scene name.

| Ratio $p >= 0.05$ | withdraw | higgs | grid | insurance | climate | alert |
|---|---|---|---|---|---|---|
| Het(Linear)-IID-ROCAUC | 0.96 | 0.93 | 0.93 | 0.93 | 1.0 | 0.97 |
| Het(Linear)-IID-ACC | 0.93 | 0.97 | 0.9 | 0.9 | 0.97 | 0.97 |
| Het(Linear)-ID-ROCAUC | 1.0 | 0.97 | 0.87 | 0.9 | 1.0 | 0.97 |
| Het(Linear)-ID-ACC | 0.93 | 0.97 | 0.87 | 0.93 | 0.93 | 1.0 |
| Het(Linear)-Bias-ROCAUC | 0.77 | 0.83 | 0.83 | 0.9 | 0.9 | 0.9 |
| Het(Linear)-Bias-ACC | 0.83 | 0.87 | 0.87 | 0.8 | 0.83 | 0.93 |
| Het(Linear)-GroupBias-ROCAUC | 0.83 | 0.88 | 0.87 | 0.88 | 0.91 | 0.87 |
| Het(Linear)-GroupBias-ACC | 0.87 | 0.89 | 0.87 | 0.88 | 0.87 | 0.89 |
| Het(MLP)-IID-ROCAUC | 0.97 | 1.0 | 0.97 | 0.97 | 0.93 | 0.93 |
| Het(MLP)-IID-ACC | 0.93 | 0.93 | 0.97 | 0.9 | 0.97 | 0.97 |
| Het(MLP)-ID-ROCAUC | 0.8 | 0.87 | 0.77 | 0.87 | 0.87 | 0.7 |
| Het(MLP)-ID-ACC | 0.8 | 0.83 | 0.83 | 0.8 | 0.73 | 0.8 |
| Het(MLP)-Bias-ROCAUC | 0.43 | 0.57 | 0.53 | 0.53 | 0.63 | 0.47 |
| Het(MLP)-Bias-ACC | 0.67 | 0.6 | 0.57 | 0.57 | 0.43 | 0.6 |
| Het(MLP)-GroupBias-ROCAUC | 0.70 | 0.76 | 0.71 | 0.73 | 0.77 | 0.69 |
| Het(MLP)-GroupBias-ACC | 0.77 | 0.73 | 0.77 | 0.74 | 0.63 | 0.73 |
| Het(SVM)-IID-ROCAUC | 0.97 | 0.97 | 0.93 | 0.97 | 0.97 | 1.0 |
| Het(SVM)-IID-ACC | 0.97 | 0.97 | 0.97 | 0.97 | 1.0 | 1.0 |
| Het(SVM)-ID-ROCAUC | 0.73 | 0.5 | 0.9 | 0.6 | 0.83 | 0.8 |
| Het(SVM)-ID-ACC | 0.9 | 0.67 | 0.37 | 0.97 | 0.97 | 0.97 |
| Het(SVM)-Bias-ROCAUC | 0.4 | 0.27 | 0.1 | 0.37 | 0.53 | 0.47 |
| Het(SVM)-Bias-ACC | 0.97 | 0.37 | 0.43 | 0.93 | 0.8 | 0.93 |
| Het(SVM)-GroupBias-ROCAUC | 0.63 | 0.48 | 0.36 | 0.52 | 0.69 | 0.66 |
| Het(SVM)-GroupBias-ACC | 0.90 | 0.56 | 0.62 | 0.90 | 0.85 | 0.86 |
| Het(RF)-IID-ROCAUC | 0.97 | 0.97 | 0.9 | 0.97 | 0.97 | 0.97 |
| Het(RF)-IID-ACC | 0.93 | 0.97 | 0.87 | 0.93 | 0.9 | 0.9 |
| Het(RF)-ID-ROCAUC | 0.97 | 0.87 | 0.9 | 0.9 | 0.83 | 0.97 |
| Het(RF)-ID-ACC | 1.0 | 0.87 | 0.9 | 0.97 | 1.0 | 0.87 |
| Het(RF)-Bias-ROCAUC | 0.73 | 0.9 | 0.77 | 0.87 | 0.93 | 0.87 |
| Het(RF)-Bias-ACC | 0.9 | 0.93 | 0.8 | 0.97 | 0.8 | 0.7 |
| Het(RF)-GroupBias-ROCAUC | 0.85 | 0.87 | 0.84 | 0.9 | 0.91 | 0.86 |
| Het(RF)-GroupBias-ACC | 0.87 | 0.89 | 0.85 | 0.89 | 0.88 | 0.85 |
| Het(LGBM)-IID-ROCAUC | 1.0 | 0.97 | 0.97 | 0.97 | 0.87 | 1.0 |
| Het(LGBM)-IID-ACC | 0.93 | 0.97 | 0.93 | 0.9 | 1.0 | 0.9 |
| Het(LGBM)-ID-ROCAUC | 1.0 | 0.87 | 0.93 | 0.93 | 0.87 | 0.93 |
| Het(LGBM)-ID-ACC | 0.97 | 0.83 | 0.97 | 1.0 | 0.97 | 0.97 |
| Het(LGBM)-Bias-ROCAUC | 0.7 | 0.93 | 0.83 | 0.73 | 0.97 | 0.9 |
| Het(LGBM)-Bias-ACC | 0.83 | 0.9 | 0.77 | 0.83 | 0.8 | 0.8 |
| Het(LGBM)-GroupBias-ROCAUC | 0.81 | 0.88 | 0.85 | 0.89 | 0.91 | 0.86 |
| Het(LGBM)-GroupBias-ACC | 0.88 | 0.88 | 0.84 | 0.90 | 0.86 | 0.85 |
| Het(XGBoost)-IID-ROCAUC | 1.0 | 0.97 | 1.0 | 0.8 | 0.93 | 1.0 |
| Het(XGBoost)-IID-ACC | 0.93 | 1.0 | 0.97 | 1.0 | 0.93 | 0.87 |
| Het(XGBoost)-ID-ROCAUC | 0.9 | 0.83 | 0.9 | 0.9 | 0.87 | 1.0 |
| Het(XGBoost)-ID-ACC | 0.97 | 0.87 | 0.93 | 0.97 | 0.93 | 0.97 |
| Het(XGBoost)-Bias-ROCAUC | 0.83 | 0.97 | 0.83 | 0.9 | 0.83 | 0.83 |
| Het(XGBoost)-Bias-ACC | 0.87 | 0.93 | 0.9 | 0.83 | 0.8 | 0.87 |
| Het(XGBoost)-GroupBias-ROCAUC | 0.88 | 0.89 | 0.90 | 0.90 | 0.87 | 0.89 |
| Het(XGBoost)-GroupBias-ACC | 0.88 | 0.89 | 0.88 | 0.88 | 0.89 | 0.88 |
| Het(CatBoost)-IID-ROCAUC | 0.93 | 0.8 | 0.9 | 0.97 | 0.93 | 0.9 |
| Het(CatBoost)-IID-ACC | 1.0 | 0.9 | 1.0 | 0.87 | 1.0 | 0.97 |
| Het(CatBoost)-ID-ROCAUC | 0.97 | 0.93 | 0.93 | 0.93 | 0.87 | 0.97 |
| Het(CatBoost)-ID-ACC | 0.97 | 0.93 | 0.83 | 0.97 | 0.97 | 0.97 |
| Het(CatBoost)-Bias-ROCAUC | 0.83 | 0.87 | 0.8 | 0.8 | 0.9 | 0.87 |
| Het(CatBoost)-Bias-ACC | 0.87 | 0.9 | 0.93 | 0.87 | 0.8 | 0.8 |
| Het(CatBoost)-GroupBias-ROCAUC | 0.86 | 0.89 | 0.88 | 0.86 | 0.89 | 0.87 |

| | | | | | | |
|---|---|---|---|---|---|---|
| Het(CatBoost)-GroupBias-ACC | 0.85 | 0.89 | 0.89 | 0.90 | 0.86 | 0.86 |

Table 7: The ratio that the p-value of test is larger than or equal to 0.05 in the 30 repeated experiments for classification base learners. The name of each row is "EvaluationModelName"-"TestName"-"MetricName". The name of each column is the scene name.

We also visualization the Q-Q plot of the baseline evaluation model in figure 6 and the assumption assessment result of our evaluation model in table 8. From the visualization, the error distribution of baseline is close to normal distribution. Although the HetEM-Linear and HetEM-CatBoost does not pass our ID check and Bias check, the dramatically empirical error reduction of our evaluation model is still *believed* as valuable if upper bound existed. Of course it can be not used to calculate the upper bound of the two evaluation models by theorem 4 directly in this scene.

| Ratio $p \geq 0.05$ | A-Share Trade |
|---|---|
| Het(Linear)-IID | 1.0 |
| Het(Linear)-ID | 0.0 |
| Het(Linear)-Bias | 0.0 |
| Het(CatBoost)-IID | 1.0 |
| Het(CatBoost)-ID | 0.0 |
| Het(CatBoost)-Bias | 0.0 |

Table 8: The ratio that p-value of test is larger than or equal to 0.05 in the 5 repeated experiments for A-share trade scene. The name of each row is "EvaluationModelName"-"TestName"-"MetricName". The name of each column is the scene name.

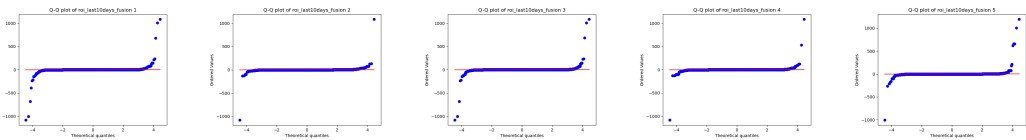

Figure 6: Q-Q plot of evaluation error of Baseline (Last10DayRoI).

# F    Hyper-parameters of evaluation model

There are 7 kinds of base learners for the evaluation model learning in our experiments. We use the default hyper-parameters for the 7 kinds of base learners in 6 scenes where evaluation metrics are RMSE and $R^2$ as following.

```
{
  "Linear Regression": {
    "copy_X": true,
    "fit_intercept": true,
    "positive": false
  },
  "MLP": {
    "activation": "relu",
    "alpha": 0.0001,
    "batch_size": "auto",
    "beta_1": 0.9,
    "beta_2": 0.999,
    "early_stopping": false,
    "epsilon": 1e-08,
    "hidden_layer_sizes": [
      100
    ],
    "learning_rate": "constant",
```

```
1006        "learning_rate_init": 0.001,
1007        "max_fun": 15000,
1008        "max_iter": 200,
1009        "momentum": 0.9,
1010        "n_iter_no_change": 10,
1011        "nesterovs_momentum": true,
1012        "power_t": 0.5,
1013        "shuffle": true,
1014        "solver": "adam",
1015        "tol": 0.0001,
1016        "validation_fraction": 0.1,
1017        "verbose": false,
1018        "warm_start": false
1019      },
1020      "SVM": {
1021        "C": 1.0,
1022        "cache_size": 200,
1023        "coef0": 0.0,
1024        "degree": 3,
1025        "epsilon": 0.1,
1026        "gamma": "scale",
1027        "kernel": "rbf",
1028        "max_iter": -1,
1029        "shrinking": true,
1030        "tol": 0.001,
1031        "verbose": false
1032      },
1033      "RF": {
1034        "bootstrap": true,
1035        "ccp_alpha": 0.0,
1036        "criterion": "squared_error",
1037        "max_features": 1.0,
1038        "min_impurity_decrease": 0.0,
1039        "min_samples_leaf": 1,
1040        "min_samples_split": 2,
1041        "min_weight_fraction_leaf": 0.0,
1042        "n_estimators": 100,
1043        "verbose": 0,
1044        "warm_start": false
1045      },
1046      "XGBoost": {
1047        "objective": "reg:squarederror",
1048        "missing": "nan",
1049        "n_estimators": 100
1050      },
1051      "LightGBM": {
1052        "boosting_type": "gbdt",
1053        "colsample_bytree": 1.0,
1054        "importance_type": "split",
1055        "learning_rate": 0.1,
1056        "max_depth": -1,
1057        "min_child_samples": 20,
1058        "min_child_weight": 0.001,
1059        "min_split_gain": 0.0,
1060        "n_estimators": 100,
1061        "num_leaves": 31,
1062        "reg_alpha": 0.0,
1063        "reg_lambda": 0.0,
1064        "subsample": 1.0,
```

```
1065        "subsample_for_bin": 200000,
1066        "subsample_freq": 0
1067      },
1068      "CatBoost": {
1069        "loss_function": "RMSE"
1070      }
1071    }
```

The default hyper-parameters of the 7 kinds of base learners in 5 scenes where evaluation metrics are
ROC-AUC and ACC as following.

```
1074    {
1075      "LogisticRegression": {
1076        "C": 1.0,
1077        "dual": false,
1078        "fit_intercept": true,
1079        "intercept_scaling": 1,
1080        "max_iter": 100,
1081        "multi_class": "deprecated",
1082        "penalty": "l2",
1083        "solver": "lbfgs",
1084        "tol": 0.0001,
1085        "verbose": 0,
1086        "warm_start": false
1087      },
1088      "MLP": {
1089        "activation": "relu",
1090        "alpha": 0.0001,
1091        "batch_size": "auto",
1092        "beta_1": 0.9,
1093        "beta_2": 0.999,
1094        "early_stopping": false,
1095        "epsilon": 1e-08,
1096        "hidden_layer_sizes": [100],
1097        "learning_rate": "constant",
1098        "learning_rate_init": 0.001,
1099        "max_fun": 15000,
1100        "max_iter": 200,
1101        "momentum": 0.9,
1102        "n_iter_no_change": 10,
1103        "nesterovs_momentum": true,
1104        "power_t": 0.5,
1105        "shuffle": true,
1106        "solver": "adam",
1107        "tol": 0.0001,
1108        "validation_fraction": 0.1,
1109        "verbose": false,
1110        "warm_start": false
1111      },
1112      "SVM": {
1113        "C": 1.0,
1114        "break_ties": false,
1115        "cache_size": 200,
1116        "coef0": 0.0,
1117        "decision_function_shape": "ovr",
1118        "degree": 3,
1119        "gamma": "scale",
1120        "kernel": "rbf",
1121        "max_iter": -1,
1122        "probability": false,
```

```
1123        "shrinking": true,
1124        "tol": 0.001,
1125        "verbose": false
1126      },
1127      "RandomForest": {
1128        "bootstrap": true,
1129        "ccp_alpha": 0.0,
1130        "criterion": "gini",
1131        "max_features": "sqrt",
1132        "min_impurity_decrease": 0.0,
1133        "min_samples_leaf": 1,
1134        "min_samples_split": 2,
1135        "min_weight_fraction_leaf": 0.0,
1136        "n_estimators": 100,
1137        "oob_score": false,
1138        "verbose": 0,
1139        "warm_start": false
1140      },
1141      "XGBoost": {
1142        "objective": "binary:logistic",
1143        "enable_categorical": false,
1144        "missing": "NaN"
1145      },
1146      "LightGBM": {
1147        "boosting_type": "gbdt",
1148        "colsample_bytree": 1.0,
1149        "importance_type": "split",
1150        "learning_rate": 0.1,
1151        "max_depth": -1,
1152        "min_child_samples": 20,
1153        "min_child_weight": 0.001,
1154        "min_split_gain": 0.0,
1155        "n_estimators": 100,
1156        "num_leaves": 31,
1157        "reg_alpha": 0.0,
1158        "reg_lambda": 0.0,
1159        "subsample": 1.0,
1160        "subsample_for_bin": 200000,
1161        "subsample_freq": 0
1162      },
1163      "CatBoost": {}
1164  }
```

1165 We also use the Grid search for the hyperparameter searching in quantum trade scene, the searched
1166 range is listed as following, the cross validation for the searching is 3-fold cross validation.

```
1167        "CatBoost" = {
1168            'iterations': [100, 200],
1169            'depth': [6, 8],
1170            'learning_rate': [0.01, 0.1],
1171            'l2_leaf_reg': [1, 3],
1172            'border_count': [32, 64],
1173        }
```

# G   Original evaluation errors

1175 We listed the original average root mean square evaluation errors for 4 different evaluation metrics:
1176 ROC-AUC, ACC, RMSE, $R^2$ in the 11 scenes for evaluation model learning. The reason we do
1177 not use the policy risk, Area Under the Qini Curve (Qini Coefficient, Radcliffe [2007]) and Area

Under the Uplift Curve (AUCC, Radcliffe [2007]) for discrete target is that not all models return a
probability for classification. For example, the output continuous value of logistic regression can not
be used to represent the probability that it belongs to a class.

| RMSE | TERECO | Calculus | kin8nm | NuScale | Ad |
|---|---|---|---|---|---|
| Holdout-100 | 0.45471 | 0.26197 | 0.00499 | 0.01521 | 0.10611 |
| Holdout-50 | 0.28538 | 0.32978 | 0.00435 | 0.01328 | 0.19199 |
| Holdout-20 | 0.49894 | 0.29417 | 0.03387 | 0.01584 | 0.21959 |
| Holdout-10 | 0.39836 | 0.22317 | 0.00591 | 0.01706 | 0.21805 |
| CV-5 | 0.43369 | 0.25795 | 0.00525 | 0.01518 | 0.07848 |
| CV-10 | 0.39857 | 0.25568 | 0.00551 | 0.01513 | 0.06187 |
| Bootstrap | 0.46825 | 0.26353 | 0.00977 | 0.01544 | 0.13219 |
| Het-Linear | 0.00474 | 0.00918 | 0.00027 | 0.00077 | 0.00555 |
| Het-MLP | 0.03528 | 0.31249 | 0.02601 | 0.05205 | 0.03862 |
| Het-SVR | 0.03881 | 0.04573 | 0.03729 | 0.02167 | 0.03665 |
| Het-RF | 0.01348 | 0.00513 | 0.00193 | 0.00335 | 0.01378 |
| Het-LGBM | 0.00998 | 0.00530 | 0.00219 | 0.00301 | 0.00985 |
| Het-XGBoost | 0.01223 | 0.00549 | 0.00235 | 0.00331 | 0.01183 |
| Het-CatBoost | 0.00756 | 0.00468 | 0.00159 | 0.00241 | 0.00695 |

Table 9: Evaluation error when evaluation metric is RMSE.

| RMSE | TERECO | Calculus | kin8nm | NuScale | Ad |
|---|---|---|---|---|---|
| Holdout-100 | 0.12591 | 0.38410 | 0.01355 | 0.00946 | 0.19670 |
| Holdout-50 | 0.11428 | 0.53892 | 0.04429 | 0.01026 | 7.52594 |
| Holdout-20 | 0.20604 | 0.74010 | 0.03152 | 0.01594 | 6.24597 |
| Holdout-10 | 0.15633 | 0.70377 | 0.04148 | 0.02123 | 5.58360 |
| CV-5 | 0.19360 | 0.42477 | 0.01200 | 0.00950 | 2.98845 |
| CV-10 | 0.25662 | 0.43089 | 0.01052 | 0.00961 | 3.72466 |
| Bootstrap | 0.27573 | 0.39105 | 0.01911 | 0.01065 | 0.24769 |
| Het-Linear | 0.01689 | 0.00061 | 0.00061 | 0.00174 | 0.01208 |
| Het-MLP | 0.04252 | 0.02690 | 0.02690 | 0.05219 | 0.04072 |
| Het-SVR | 0.06077 | 0.04628 | 0.04628 | 0.03578 | 0.05020 |
| Het-RF | 0.04748 | 0.00441 | 0.00441 | 0.00770 | 0.03020 |
| Het-LGBM | 0.03539 | 0.00508 | 0.00508 | 0.00702 | 0.02145 |
| Het-XGBoost | 0.04258 | 0.00521 | 0.00521 | 0.00756 | 0.02537 |
| Het-CatBoost | 0.02669 | 0.00368 | 0.00368 | 0.00553 | 0.01515 |

Table 10: Evaluation error when evaluation metric is $R^2$.

| RMSE | Withdraw | Higgs | Grid | Insurance | Climate | Alert |
|---|---|---|---|---|---|---|
| Holdout-100 | 0.02443 | 0.02132 | 0.01162 | 0.04982 | 0.06820 | 0.02185 |
| Holdout-50 | 0.03003 | 0.02323 | 0.01125 | 0.08180 | 0.07170 | 0.02442 |
| Holdout-20 | 0.03636 | 0.02878 | 0.01143 | 0.08537 | 0.07966 | 0.02515 |
| Holdout-10 | 0.04218 | 0.04217 | 0.01355 | 0.09063 | 0.12213 | 0.03166 |
| CV-5 | 0.02445 | 0.02115 | 0.01161 | 0.06179 | 0.06894 | 0.02144 |
| CV-10 | 0.02435 | 0.02118 | 0.01164 | 0.06180 | 0.06924 | 0.02159 |
| Bootstrap | 0.02634 | 0.02335 | 0.01236 | 0.04996 | 0.07509 | 0.02387 |
| Het-Linear | 0.01587 | 0.01438 | 0.00723 | 0.01148 | 0.05123 | 0.01391 |
| Het-MLP | 0.04397 | 0.04000 | 0.03974 | 0.05053 | 0.06654 | 0.04581 |
| Het-SVR | 0.03433 | 0.02250 | 0.05526 | 0.03135 | 0.06141 | 0.07091 |
| Het-RF | 0.02107 | 0.01608 | 0.01014 | 0.01645 | 0.05892 | 0.02255 |
| Het-LGBM | 0.01934 | 0.01517 | 0.01056 | 0.01407 | 0.05555 | 0.01890 |
| Het-XGBoost | 0.02116 | 0.01635 | 0.01016 | 0.01595 | 0.05970 | 0.02229 |
| Het-CatBoost | 0.01748 | 0.01432 | 0.00888 | 0.01262 | 0.05232 | 0.01839 |

| RMSE | Withdraw | Higgs | Grid | Insurance | Climate | Alert |
|---|---|---|---|---|---|---|
| Holdout-100 | 0.05444 | 0.03244 | 0.01145 | 0.17974 | 0.10162 | 0.02764 |
| Holdout-50 | 0.06632 | 0.02550 | 0.01119 | 0.22168 | 0.13740 | 0.03271 |
| Holdout-20 | 0.06731 | 0.03507 | 0.01056 | 0.26858 | 0.16551 | 0.03930 |
| Holdout-10 | 0.06710 | 0.04374 | 0.01312 | 0.28142 | 0.19290 | 0.03623 |
| CV-5 | 0.05443 | 0.03244 | 0.01145 | 0.17974 | 0.10145 | 0.02764 |
| CV-10 | 0.05443 | 0.03244 | 0.01145 | 0.17973 | 0.10135 | 0.02765 |
| Bootstrap | 0.05489 | 0.03481 | 0.01266 | 0.17976 | 0.10362 | 0.02825 |
| Het-Linear | 0.01850 | 0.01438 | 0.00696 | 0.01887 | 0.03357 | 0.01391 |
| Het-MLP | 0.04886 | 0.03994 | 0.03924 | 0.05166 | 0.06719 | 0.04581 |
| Het-SVR | 0.07261 | 0.03029 | 0.03749 | 0.07211 | 0.06634 | 0.07091 |
| Het-RF | 0.03444 | 0.01641 | 0.00973 | 0.03068 | 0.05216 | 0.02255 |
| Het-LGBM | 0.02929 | 0.01526 | 0.00972 | 0.02610 | 0.04359 | 0.02255 |
| Het-XGBoost | 0.03395 | 0.01656 | 0.01000 | 0.02981 | 0.05219 | 0.01890 |
| Het-CatBoost | 0.02655 | 0.01443 | 0.00879 | 0.02426 | 0.04645 | 0.01839 |

Table 12: Evaluation error when evaluation metric is ACC.

## H Evaluation cost

We list the table to compare the average evaluation time of experimental (simulation-based) evaluation and our computation evaluation as following table 13. We use the open-source data in table 13 to build the benchmark to test the computational evaluation methods.

| Scenes | Experiment (Simulation) Time | Computation Time |
|---|---|---|
| Alert (Wilson et al. [2021]) | 1 year, 8 months, and 15 days | 0.176s |
| Withdraw (Wilson et al. [2023]) | 1 year, 3 months, and 13 days | 0.077s |
| Grid-S (Schäfer et al. [2016]) | 1500s | 0.198s |
| Higgs-S (Baldi et al. [2014]) | (expected) 15 hours | 0.151s |
| Insurance (Kumar [2023]) | >1 year | 0.500s |
| Climate-S (Lucas et al. [2013]) | unknown | 0.132s |
| TERECO (Li et al. [2022]) | 34 weeks | 0.014s |
| Calculus (Kramer et al. [2023]) | 3 semesters | 0.035s |
| Ad (Gokagglers [2018]) | 3 campaigns | 0.034s |
| kin8nm-S (Fischer [2022]) | unknown | 0.038s |
| NuScale-S (Huu Tiep [2024]) | 1600 hours | 0.046s |
| A-Share Trade | 10 days | 0.103s |

Table 13: Expected evaluation time comparison in 12 scenes per agent. "S" means simulation.

## I Upper bounds tables

We list the upper bound of generalized evaluation error and generalized causal effect evaluation error in the table 14 and table 15 for researchers' quick query before data collection and upper bound estimation after learning.

## J Paradigm of computational evaluation

In order to help user to better understand the paradigm of computational evaluation, we plot the decision tree to help evaluatology researchers conduct their own research as shown in the figure 7.

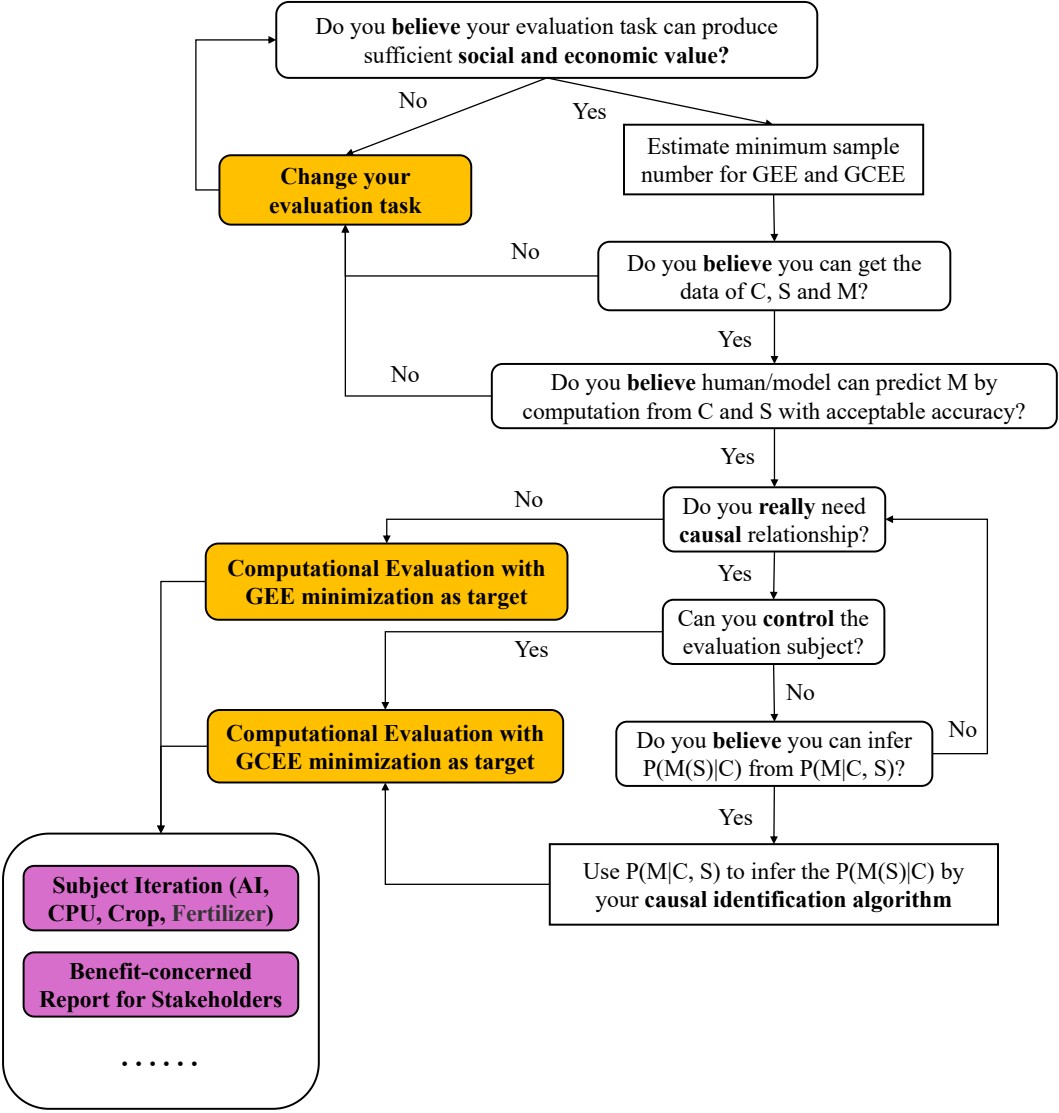

Figure 7: The decision tree that how to use computational evaluation models in your own evaluation task. The GEE means generalized evaluation error and the GCEE means generalized causal evaluation error. C is evaluation condition, S is evaluation subject, and M is evaluation metric. $P(M(S)|EC)$ is the potential outcome of evaluation metric after deploying the subject given evaluation condition.

| Samples/Confidence | 0.5 | 0.95 | 0.99 | 0.999 |
|---|---|---|---|---|
| 10 | $E + 0.186$ | $E + 0.387$ | $E + 0.480$ | $E + 0.588$ |
| 20 | $E + 0.132$ | $E + 0.274$ | $E + 0.339$ | $E + 0.416$ |
| 30 | $E + 0.107$ | $E + 0.223$ | $E + 0.277$ | $E + 0.339$ |
| 100 | $E + 0.059$ | $E + 0.122$ | $E + 0.152$ | $E + 0.186$ |
| 1000 | $E + 0.0186$ | $E + 0.0387$ | $E + 0.0480$ | $E + 0.0588$ |
| 10000 | $E + 0.00589$ | $E + 0.0122$ | $E + 0.0151$ | $E + 0.0186$ |
| 100000 | $E + 0.00186$ | $E + 0.00387$ | $E + 0.00480$ | $E + 0.00588$ |
| 1000000 | $E + 0.000589$ | $E + 0.00122$ | $E + 0.00152$ | $E + 0.00186$ |
| 10000000 | $E + 0.000186$ | $E + 0.000387$ | $E + 0.000480$ | $E + 0.000588$ |
| 100000000 | $E + 0.0000589$ | $E + 0.000122$ | $E + 0.000152$ | $E + 0.000186$ |
| 1000000000 | $E + 0.0000186$ | $E + 0.0000387$ | $E + 0.0000480$ | $E + 0.0000588$ |

Table 14: Upper bound of normalized generalized error. E is the empirical prediction error.

| Samples/Confidence | 0.5 | 0.95 | 0.99 | 0.999 |
|---|---|---|---|---|
| 10 | $2E + 0.372$ | $2E + 0.774$ | $2E + 0.960$ | $2E + 1.18$ |
| 20 | $2E + 0.263$ | $2E + 0.547$ | $2E + 0.678$ | $2E + 0.831$ |
| 30 | $2E + 0.215$ | $2E + 0.447$ | $2E + 0.554$ | $2E + 0.679$ |
| 100 | $2E + 0.118$ | $2E + 0.245$ | $2E + 0.303$ | $2E + 0.372$ |
| 1000 | $2E + 0.0372$ | $2E + 0.0774$ | $2E + 0.0960$ | $2E + 0.118$ |
| 10000 | $2E + 0.0118$ | $2E + 0.0245$ | $2E + 0.0303$ | $2E + 0.0372$ |
| 100000 | $2E + 0.00372$ | $2E + 0.00774$ | $2E + 0.00960$ | $2E + 0.0118$ |
| 1000000 | $2E + 0.00118$ | $2E + 0.00245$ | $2E + 0.00303$ | $2E + 0.00372$ |
| 10000000 | $2E + 0.000372$ | $2E + 0.000774$ | $2E + 0.000960$ | $2E + 0.00118$ |
| 100000000 | $2E + 0.000118$ | $2E + 0.000245$ | $2E + 0.000303$ | $2E + 0.000372$ |
| 1000000000 | $2E + 0.0000372$ | $2E + 0.0000774$ | $2E + 0.0000960$ | $2E + 0.000118$ |

Table 15: Upper bound of normalized generalized causal effect error. E is the empirical prediction error.

