# OpenReview forum: "A Computational Theory for Efficient Mini Agent Evaluation with Causal Guarantees"
_NeurIPS.cc/2025/Conference — Submitted to NeurIPS 2025_

### Official Review · Reviewer_gSDJ · 2025-06-01

**Clarity:** 1
**Significance:** 1
**Originality:** 2
**Rating:** 1
**Confidence:** 5

**Summary:**

The paper proposes a “computational evaluation” framework in which a learned meta-model predicts how an agent will score on downstream tasks, with theoretical upper bounds on generalization and “causal-effect” errors. Experiments span 12 heterogeneous scenes (medicine, climate simulation, insurance, advertising, quantitative trading, etc.) and claim large error reductions (24 %–99 %) and up to 10⁷ × evaluation-time speed-ups compared with simple validation procedures. The authors recommend statistical tests to check key assumptions and acknowledge several limitations.

**Questions:**

- **Assumption robustness.** Can you empirically demonstrate performance when IRIS/IIDE is violated (e.g. temporally-correlated or adversarially-generated agents)?
- **Baseline selection.** Why were modern offline-policy-evaluation or doubly-robust estimators (e.g. MAGI, DR-OPE) excluded? Please add them or justify clearly.
- **Task provenance & scaling.** Several datasets are synthetic or tiled 1 600×. Could you release raw data splits and show results on a public benchmark (e.g. RL-Unplugged) without artificial inflation?
- **Causal validity.** Theorem 2 labels a mean-squared bound as “strict causal advantage.” What causal identification conditions are actually used, and how do you satisfy them in practice?
- **Reproducibility.** Will you publish code, hyper-parameters and wall-clock logs?

**Ethical Concerns:**

["NO or VERY MINOR ethics concerns only"]

**Final Justification:**

After reviewing both the authors’ response and the Final Remarks, my judgment remains unchanged. For example, the absence of a Related Work section is not addressed at all. The authors should take the paper’s clarity and presentation more seriously.

**Limitations:**

Partially addressed. The paper concedes assumption fragility and scalability gaps, but does NOT discuss:

(i) potential mis-estimation harms when evaluation models are deployed in safety-critical domains (medicine, trading);

(ii) environmental cost of training on inflated ×1 600 datasets;

(iii) how causal-effect mis-specification could bias decision-making.

Please expand the societal-impact discussion to cover these points.

**Paper Formatting Concerns:**

- Exceeds 9-page limit (core proofs and figures pushed to appendix).
- Missing "Related Work" section.
- Figure numbering resets (e.g. “Figure 1” referred to as “Figure 7” in text).
- Typos such as “addressing on on” and inconsistent capitalization of sections.
- Citations sometimes appear as raw BibTeX keys instead of formatted references.

**Quality:**

1

**Strengths And Weaknesses:**

# Strength
- **Breadth of evaluation.** The method is tested on 12 distinct scenes, ranging from medical RCT data (Alert, Withdraw) to scientific simulations, advertising click-logs and a “quantum trade” task, which demonstrates domain ambition and scalability.
- **Large empirical gains.** Reported error reductions reach 24 %–99 % across metrics and scenes, with acceleration ratios between 1,000 × and 10,000,000 × compared with experiment-or simulation-based evaluation.
- **Formal error bounds.** Theorem 1 gives a Hoeffding-style bound on generalized evaluation error; Theorems 2–4 extend this to first-order “causal effect” error under stated assumptions.
- **Meta-learning design.** A clear engineering contribution is the proxy-metric + heterogeneous-vectorization pipeline with multiple base learners (Linear, RF, CatBoost, etc.) and automatic model selection.
- **Attempted assumption testing.** The paper outlines concrete statistical tests for IIDE (Kolmogorov–Smirnov, D’Agostino-Pearson) and unbiasedness (t-tests) to falsify its own premises.
- **Self-reported limitations.** Section 7 openly notes that assumptions may fail and that LLM-scale agents are future work.

# Weakness
- **Writing and presentation.** Numerous grammatical issues (e.g., “addressing on on”), inconsistent figure numbering (e.g., Figure 1 → 7), and overuse of acronyms hinder readability.
- **Page-limit violation & appendix dependence.** Core proofs, datasets, hyper-parameters and even the workflow decision tree appear only in the appendix, while the main nine pages are dominated by notation and restated inequalities.
- **Weak baselines & reporting.** Figure 3 compares only against trivial validation splits; state-of-the-art OPE or doubly-robust estimators are ignored, and no confidence intervals on claimed 99 % gains are supplied.
- **Unrealistic assumptions & reproducibility concerns.** Guarantees require every agent to be *independently, randomly, identically sampled* (IRIS) and all errors IID (IIDE)—assumptions violated in most real-world settings. The proposed tests are merely necessary, not sufficient. No code or detailed protocol is provided; running ~100 M samples on a single laptop in two days seems implausible.
- **Arbitrary, inflated task suite.** Several scenes are synthetic or Kaggle datasets; sample sizes are inflated by tiling each record 1,600×, yielding 14 M “training” examples for *Insurance* on a MacBook M1. Task motivation and provenance are unclear.
- **Causal claims not justified.** Theorem 2’s “strict causal advantage” simply restates MSE properties under the IRIS/unbiased assumptions, without an identification strategy; causal language is substantially overstated.
- **Over-claiming impact & lack of survey.** The paper asserts ethical safeguards and applicability to “infinite agent spaces,” yet empirical evidence remains toy-scale. The manuscript omits a structured survey of prior art in offline-policy evaluation, PAC learning, causal OPE, etc., and compares only with naïve hold-out and bootstrap baselines.

---

> ### Author Rebuttal · Authors · 2025-07-30
>
> Some misunderstandings
>
> At the begin of the responses, we humbly remarked that our work is mainly for the meta-evaluation task (evaluate the evaluation approach) rather than causal task. The purpose of this work is to reduce the evaluation cost while keep acceptable evaluation errors rather than to learn a causal model for general causal tasks, although we add causal insights in our evaluation model. We respectively mentioned that please do not to generalize and compare it for general causal tasks which is beyond the range of our work.
>
> Following the quote of Box, ”all models are wrong, but some are useful”, the casual property is not the first requirement of in our computational evaluation although it is one of the most important features. The first requirement is to find a useful evaluation model to reduce the evaluation costs while keep acceptable evaluation errors. Someone may do not want the evaluation is causal, that’s OK, while our computational methods also work for those requirements. It is very useful because the evaluation approach for mini agents was applied in those 6 real scenes in our experiments: “improve the mortality by individualized pop-up box”, “improve the student’s calculus grade by individualized classes”, “improve the Insurance and AD efficiency by individualized recommendation decision”, “improve the RoI by decision of mini agents for trade”.
>
> Of course, if there are any other related works from reviewers that may be useful for the computational evaluation task, we’d like to add an extra discussion part about them.
>
> 1.Assumption robustness
>
> QUESTION: Can you empirically …… or adversarially-generated agents)?
>
> RESPONSE: That’s really a good question. Our bound are not very tight so that it is robust in the following cases.
>
> (a)	IIDE assumption
>
> 1)How to satisfy it and why it is useful?
>
> We remark that all the error is hoped as IID by researchers. If it is not IIDE, the best choice is not to test its robustness to increase the confidence of the assumptions but to change the metric prediction model to make the errors unpredictable (boosting algorithms can make it unpredictable for given functions), independently, and randomly. It gives us a SECOND CHANCE to satisfy the assumption if the (s,c,m) is not IID. This is the main difference in theory between our work and other similar works (PAC, DR, other offline-policy in line 160-166) based on the IID assumption.
>
> 2)empirical non-IIDE case study
>
> Because the upper bounds cannot be measured. And we cannot list all the cases that violate the IIDE assumption. So, we design 3 synthetic experiment to show the robustness of the IIDE. The first non-IIDE situation is distribution shifting of errors. The second non-IIDE situation is error dependency (temporal). The third non-IIDE situation is the outliers (black swan events). We use the three situations to test the robustness of our bounds.
>
> The condition is N(0,1). The mini agent assignment is N(0,1). The noise is N(0,0.1). The evaluation metric function f(c,s,e) is np.clip(0.3 * c + 0.5 * s + e, 0, 1). The evaluation metric predictive model is np.clip(0.3 * c + 0.5 * s, 0, 1). The number of samples is 500.
>
> Here is the robustness test experiment result:
>
> Name |Our Theoretical Bound | IIDE | non-IIDE1 | non-IIDE 2| non-IIDE 3|
>
> Error Bound |0.06484732406369578 | 0.010114041012576054| 0.03348055885517107| 0.039488422074875774| 0.02999711253327391|
>
> Individual Causal Effect Error Bound |0.1096772717167341 |0.00010535280724731346|0.08964805476948194| 0.10440750588976608| 0.08760353747620356|
>
> From the experiments, we can see that even in those non-IID cases, our evaluation error bound, and causal evaluation error bound are still worked.
>
> (b)	IRIS assumption
>
> 1)When to check it?
>
> First, IRIS assumption is not necessary if the researcher does not need the evaluation with causal guarantee.
>
> 2)How to satisfy it?
>
> Second, the mini agents are randomly deployed in real scenes. So, the IRIS assumption is always satisfied in our experiments. Before we collect the data, we should randomly deploy mini agents to make the IRIS to be true. There are no unobserved confounders between the mini agents and the outcome metrics in our tasks, so “the IRIS is not satisfied” is no exist in our works.
>
> 3)The central role of IRIS assumption for causal property of evaluation model
>
> Third, the causal property is from the data generating process before the data collecting, rather than from any kinds of data analysis approach or estimators, such as propensity score, double robust estimators, or other mentioned deep-learning networks, such as TARNet and GANITE. If the IRIS assumption is violated, we cannot exclude the existence of “hidden common cause which value is identical with the treatment assignment” without new STRONG assumptions. As said as Fisher “it would be impossible to present an exhaustive list of such possible differences appropriate to any kind of experiment, because the uncontrolled causes that may influence the result are always strictly innumerable”.
>
> 4)emperical non-IRIS case study
>
> For example, for temporally correlated sampled agents and adversarial-generated agents that violate the IRIS, the individual causal effect from the mini agents to the metrics, is not identifiable without new assumptions due to the HICC. It is aganist our initial idea to simply the assumptions as much as we can in computational evaluation. Those works can be found in Time-series Deconfounders, GANITE as pointed in line 967-972 of appendix E and line 114-119 of section 3.2. Those assumptions including IID (IID s, IID c, and IID m) are regarded as too STRONG for the computational evaluation task with causal guarantee.
>
> The bound is also worked for temporally correlated agents and adversarial-generated agents, we recommend using Markov process, and Game theory for detailed analysis. Here we give an empirical result:
>
> The sample number is 500 with 95% confidence. The c is sampled from N(0,1). The e is sampled from N(0,1). For temporally-correlated agents, it is generated by AR(1) to simulate the temporal correlation. For adversarially-generaed agents, it is generated by alternately pushed the agents towards extreme values (0.05 and 0.95) to simulate the worst-case scenario of adversarial selection.
>
> Here is the empirically experiments:
>
> Name |Our Theoretical Bound | Temporally-Correlated Agents | Adversarially-generated Agents |
>
> MSE | 0.003688879454113936 | 0.007143852947639841 | 2.435916193660976e-33 |
>
> Causal Effect MSE | 0.12147229238166103 | 2.1678267454022727e-33 | ~0.0|
>
> From the experiments, we can see that even in those non-IRIS cases, our evaluation error bound, and causal evaluation error bound are still worked.
>
> (c)	UNBIASED assumption
>
> 1)How to satisfy it?
>
> We hope the prediction model is unbiased. If it is not, then we can adjust our model and minus the empirical bias from the current data to increase our confidence that it is unbiased.
>
> 2)How robustness it is?
>
> It is trivial to extend the bound in theorem 1 and theorem 4 to prediction model with bias as pointed in the line 159:
>
> Evaluation Error Up Bound: $E_{emp}(\hat{f})+\sqrt{\frac{1}{2n}\log(\frac{1}{\sigma})}-B)$
>
> Causal Effect Error Up Bound: $ 2(E_{emp}(\hat{f})+\sqrt{\frac{1}{2n}\log(\frac{1}{\sigma})}-B))$
>
> Where B is the metric prediction bias $E(\epsilon)$. The minus term does not mean it would be smaller than unbiased cases. Because the bias will also increase the $E_{emp}(\hat{f})$.
>
> For Cauchy-like distribution without bias, it will not be discussed because it is usually a special counterexample theoretically rather than a real case.
>
> Baseline selection problem.
>
> As pointed in the line 967-972, we did a detailed survey about the ‘’SOTA causal inference methods’’. Actually, the SOTA approach is CatBoost tree-like methods to handling the data we used rather than the mentioned approaches (except the TabPFN which need huge computing). However, we’d like to add discussion why each of them is not included.
>
> Task provenance & scaling.
>
> QUESTION: Several datasets are synthetic …… (e.g. RL-Unplugged) without artificial inflation?
>
> RESPONSE:
> The 1600 is the number of mini agents in the training data. We need to know the pair result for each unit. The typo in table 5 is ‘0.8’ in train samples, rather than ‘0.2’. It assumes that we use the binary metric in the table. However, if we use a more high-level metric, such as ACC. The sample number would be 1600. That depended on the requirements. We respectively pointed that the procedure of true evaluation metric generation in appendix D.1 (evaluation model) and D.2 (conditional evaluation model).
>
> QUESTION: Show results on a public benchmark (e.g. RL-Unplugged)
>
> RESPONSE: The dataset of our proposed benchmark is public from real experiments with high-economic values, such as ICU, Insurance, Calculus Teaching, and Trade. While other benchmarks may not satisfy our need for the meta-evaluation task: IRIS assumption has been satisfied in our benchmark in all 6 scenes with real randomized experiments. The RL-Unplugged is a very good simulation scene for computational evaluation task, we will develop more useful computational evaluation strategies for this simulation scene and compared with SOTA in this field in future.
>
> Causal validity.
>
> QUESTION: Theorem 2 labels …… and how do you satisfy them in practice?
>
> RESPONSE:
>
> Identification conditions: The assumptions in the theorem 2 are IRIS of the mini agents, and compared causal models are unbiased without any other assumptions. The IRIS can be satisfied by randomly deployed the mini agents in the given space. The unbias assumption can be satisfied by adjust the metric prediction model.
>
> Reproducibility.
>
> The code, hyperparameters, has been uploaded in the appendix. The wall-clock logs files, our built benchmark will be publish on the open-accessed repository.
>
> Paper Formatting Concerns.
>
> The proposed concerns have been addressed in our revised version.

---

> ### Author Response · Authors · 2025-08-04
>
> Given the character limit, we include a more detailed explanation in the response below.
>
> ### **Further Clarification on Causal Validity**
>
> **On Identification Assumptions:**
> Causal identification relies on assumptions—not merely on the data. Whether expressed via conditional independence, unconfoundedness, or structural causal models (SCM), these assumptions are essential for inferring causal effects. Canonical methods such as the ID and IDC algorithms of Ilya Shpitser (*Complete Identification Algorithm*, proposed in 2006), or some of Tian's work (proposed in 2002) assume fully specified SCMs and i.i.d. sampling across units (including condition, subject, and metrics for meta-evaluation task).
>
> Our framework differs in both purpose and scope. We do not aim to learn causal effects directly; rather, we propose a computational evaluation approach that provides *provable causal guarantees under milder assumptions*. In particular, our Theorem 2 offers a theoretical bridge between generalized evaluation error and generalized effect evaluation error for infinite mini agents. Since generalized errors are unobservable in real applications, Theorems 3 and 4 operationalize this connection for use in practice.
>
> A key innovation lies in the IRIS assumption, which—unlike traditional causal inference approaches—allows for the **condition variables and evaluation metrics to be non-i.i.d.** This setting is more aligned with real-world agent deployment scenarios and is strictly more general than the assumptions made in doubly robust estimators or other estimators with IID assumption.
>
> We have also provided detailed implementation procedures for testing and mitigating assumption violations (IIDE, IRIS, and unbiasedness), as described in our earlier response on identification robustness. The framework remains valid and bounded even under adversarial and temporally correlated settings, as shown in our empirical results.
>
> ---
>
> ### **Clarification on Formatting and Presentation**
>
> **❑ Exceeds 9-page limit**
> Our main submission adheres strictly to the NeurIPS 9-page limit. All supplementary proofs and experimental details are placed in the appendix, as allowed. If the reviewer believes that certain material is critical for understanding, we are happy to move it into the main body using the optional extra page.
>
> **❑ Missing “Related Work” section**
> Due to space constraints, related theoretical work and meta-evaluation literature are discussed contextually throughout the paper. A more structured summary is provided in Appendices B, D, and E. We welcome specific suggestions and are prepared to add a focused related work section in the final version if needed.
>
> **❑ Figure numbering / typos**
> Minor issues such as inconsistent figure numbering and typographical errors (“addressing on on”) have been corrected. We appreciate the reviewer’s attention to detail.
>
> **❑ Citation style**
> We use the author-year citation format provided by the official NeurIPS template. If numeric citations are preferred, we will adopt them in the final version.
>
> ---
>
> ### **Conclusion and Request for Reconsideration**
>
> We respectfully emphasize that this work introduces a novel, theoretically grounded framework for scalable agent evaluation with causal guarantees—validated across 12 real-world domains. It is designed to address fundamental challenges in evaluating heterogeneous agents under realistic conditions. While we welcome feedback on presentation details, we believe the core contributions and results merit strong consideration.
>
> We sincerely hope these clarifications address the concerns and lead to a more favorable evaluation.

---

> ### Author Response · Authors · 2025-08-08
>
> Dear Reviewer gSDJ,
>
> We sincerely appreciate your time and detailed feedback. We have taken your concerns very seriously and made substantial revisions in response to every point you raised. In particular, we addressed your comments on "assumption robustness", "baseline selection,"  "the utility of tasks", "causal property", "Reproducibility and formats" and we have added new experiments and clarifications as requested.
>
> Given the extent of the changes and the high confidence (5/5) expressed in your original review, we respectfully ask whether the current manuscript, in its improved form, still warrants the lowest possible score (1/6). We hope you might consider whether a revised evaluation would better reflect the updated quality and contributions of the work.
>
> We are, of course, open to any further revisions or clarifications you may suggest.
>
> With appreciation,
> The Authors

---

### Official Review · Reviewer_Mu2b · 2025-07-02

**Clarity:** 2
**Significance:** 3
**Originality:** 3
**Rating:** 3
**Confidence:** 5

**Summary:**

This paper proposes a computationally efficient framework for evaluating mini agents based on computational theory. It accelerates experimental evaluation procedures by constructing evaluation models and provides theoretical upper bounds for generalized error and causal effect error in infinite agent spaces. The paper combines theoretical analysis with empirical validation, demonstrating the effectiveness of its approach across multiple benchmark tasks (e.g., kin8nm, NuScale). Compared to traditional validation methods (e.g., Holdout, cross-validation), it achieves significant error reduction (up to 99%) in metrics such as RMSE and ROC-AUC.

**Questions:**

--Discuss the impact of assumption violations (non-IIDE/IRIS) on error bounds.
--Compare with state-of-the-art causal inference methods.
--Include appropriate extension experiments on high-dimensional settings (e.g., small-scale LLMs or vision models).
--Provide more implementation details.

**Ethical Concerns:**

["NO or VERY MINOR ethics concerns only"]

**Final Justification:**

Thank you for the authors' response, which addressed some of my concerns. However, the paper still lacks theoretical analysis on robustness and parameter sensitivity. In its current form, the work does not meet NeurIPS's acceptance criteria. Therefore, I maintain my original rating.

**Limitations:**

Yes.

**Quality:**

2

**Strengths And Weaknesses:**

Strengths:
--Proposes a meta-learning framework to address evaluation challenges in heterogeneous agent spaces.
--Introduces a proxy module to enhance model generalization.
--Achieves interpretability of module contributions through Shapley values.

Weaknesses:
--Do the "boundary conditions" of the theoretical contributions overly rely on ideal assumptions?
The paper claims to provide "upper bound proofs" (e.g., Theorems 1 and 4) for generalized and causal effect errors of infinite agents via computational theory. However, these derivations depend on assumptions such as independent and identically distributed errors (IIDE) and no hidden confounding factors. In real-world agent evaluation scenarios (e.g., clinical trials or financial risk control), there are often distribution shifts, temporal dependencies, or unobserved confounders (e.g., patient genetic differences, black swan events in markets). If the theoretical bounds fail due to violated assumptions in practice, how can the framework's practicality be justified? Could robustness analysis (e.g., upper bounds under adversarial perturbations) or sensitivity analysis (e.g., tolerance to hidden confounding factors) be introduced to enhance the generality of the theory?
--Does the experimental design overlook state-of-the-art methods in causal inference?
The experiments only compare with traditional validation methods (e.g., Holdout, cross-validation), but do not engage with recent tools from the causal inference field—such as deep causal networks or counterfactual modeling. For instance, toolkits like YLearn and DoWhy-gcm have been used for high-dimensional causal effect estimation, yet the paper chooses not to include deep learning baselines. Is the superiority of the "meta-learner" approach on low-dimensional structured data merely a result of weak baseline comparisons? If directly competing with SOTA causal inference methods (e.g., TARNet, GANITE), would the performance advantage still hold?
--Lack of empirical support for generalization to "high-dimensional agents"?
The paper claims the framework is applicable to "high-dimensional agents" (e.g., large language models), but experiments are conducted only on low-dimensional datasets (kin8nm, Insurance), without discussing how parameter dimensions affect the error bounds. For example, Theorem 2 mentions "random sampling excludes hidden common causes," but the complexity of high-dimensional parameter spaces may lead to the curse of dimensionality and overfitting risks. How can the validity of the theoretical bounds and the meta-learning framework be demonstrated for agents with millions of parameters (e.g., LLMs)? Could high-dimensional synthetic data (e.g., simulated language tasks) or sparse regularization strategies (e.g., L1/L2 constraints) be used to verify scalability?
--Missing pseudocode for the meta-learning component (Section 4):
Key steps such as "vectorization strategies" are only conceptually described, which undermines reproducibility.

---

> ### Author Rebuttal · Authors · 2025-07-30
>
> Thanks for the reviewer’s positive comments. Here are the point-to-point responses.
>
> -- Discuss the impact of assumption violations (non-IIDE/IRIS) on error bounds.
>
> QUESTION: If the theoretical bounds fail due to violated assumptions in practice, how can the framework's practicality be justified?
>
> RESPONSE: It’s a good question.
>
> (a)	IIDE assumption
>
> We remark that all the error is hoped as IID by researchers. If it is not IIDE, the best choice is not to test its robustness to increase the confidence of the assumptions but to change the metric prediction model to make the errors unpredictable (boosting algorithms can make it unpredictable for given functions), independently, and randomly. It gives us a SECOND CHANCE to satisfy the assumption after (s,c,m) is non-IID generated.
>
> QUESTION: In real-world agent evaluation …… patient genetic differences, black swan events in markets).
>
> RESPONSE:
>
> Because the upper bounds cannot be measured in reality. And we cannot list all the cases that violate the IIDE assumption. So, we design 3 synthetic experiment to show the robustness of the IIDE. The first non-IIDE situation is distribution shifting of errors. The second non-IIDE situation is error dependency (temporal). The third non-IIDE situation is the outliers (black swan events). We use the three situations to test the robustness of our bounds.
>
> The condition is N(0,1). The mini agent assignment is N(0,1). The noise is N(0,0.1). The evaluation metric function f(c,s,e) is np.clip(0.3 * c + 0.5 * s + e, 0, 1). The evaluation metric predictive model is np.clip(0.3 * c + 0.5 * s, 0, 1). The number of samples is 500.
>
> Here is the robustness test experiment result:
>
> Name |Our Theoretical Bound | IIDE | non-IIDE1 | non-IIDE 2| non-IIDE 3|
>
> Error Bound |0.06484732406369578 | 0.010114041012576054| 0.03348055885517107| 0.039488422074875774| 0.02999711253327391|
>
> Individual Causal Effect Error Bound |0.1096772717167341 |0.00010535280724731346|0.08964805476948194| 0.10440750588976608| 0.08760353747620356|
>
> From the experiments, we can see that even in those non-IID cases, our evaluation error bound, and causal evaluation error bound are still worked. The reason is that our bound are not very tight so that it is robust in those cases.
>
> (b)	IRIS assumption
>
> First, IRIS assumption is not necessary if the researcher does not need the causal evaluation.
>
> Second, the mini agents are randomly deployed in real scenes. So, the IRIS assumption is always satisfied in our experiments. Before we collect the data, we should also collect the data with randomly deployed mini agents to make the IRIS to be true. There are no unobserved confounders between the mini agents and the outcome metrics (including patient genetic differences, black swan events in markets), so this concern is no exist.
>
> Third, if the IRIS assumption is violated, we cannot exclude the existence of “hidden common cause which value is IDENTICAL with the treatment assignment”. It is NEVER identifiable in any existed causal inference methods because they are IDENTICAL in measurements. It can help us to understand why the IRIS is necessary for causal evaluation as pointed in the line 109-119 in section 3.2. For example, someone may want to misleading others that he is the god of sun by synchronize with the activity (sunrise) or feature of the sun (red), or any other planets, like a rooster or red-crowned crane. And the decision in Monty Hall Problem depends on whether you believe the extra door is randomly opened or deliberately opened. Such hidden IDENTICAL common cause cannot be excluded without randomized interventions.
>
> (c)	UNBIASED assumption
>
> First, the prediction model is hoped as unbiased. If it is not, then we can adjust our model and minus the empirical bias from the current data to increase our confidence that it is unbiased.
>
> Second, it is trivial to extend the bound in theorem 1 and theorem 4 to prediction model with bias as pointed in the line 159:
>
> Evaluation Error Up Bound: $E_{emp}(\hat{f})+\sqrt{\frac{1}{2n}\log(\frac{1}{\sigma})}-B)$
>
> Causal Effect Error Up Bound: $ 2(E_{emp}(\hat{f})+\sqrt{\frac{1}{2n}\log(\frac{1}{\sigma})}-B))$
>
> Where B is the metric prediction bias $E(\epsilon)$. The minus term does not mean it would be smaller than unbiased cases. Because the bias will also increase the $E_{emp}(\hat{f})$.
>
> For Cauchy-like distribution without bias, it will not be discussed in this paper because it is usually regarded as a special counterexample theoretically rather than a real case in the world.
>
> --Compare with state-of-the-art causal inference methods.
>
> QUESTION: “Does the experimental design overlook state-of-the-art methods in causal inference? …… not to include deep learning baselines.”
>
> RESPONSE: Our work is mainly for the meta-evaluation task (to evaluate the evaluation approach) rather than causal task. As pointed in the line 842-853 of appendix B.3, we did a detailed survey about those toolkits. We decided to not include them after the careful consideration due to the different addressing research problems. For example, causal discovery is used to find a data generating process structure. The causal effect estimation applied non-observable counterfactuals. They are not comparable with our work. However, we’d like to add discussion why each of them should not be included.
>
> QUESTION: Is the superiority of the "meta-learner" approach ……, would the performance advantage still hold?”
>
> RESPONSE: As pointed in the line 967-972, we did a detailed survey about the ‘’SOTA causal inference methods’’. They are not in the same competing track. There are no others approaches about causal computational evaluation as we know due to a lack of theory to provide confidence with acceptable weak assumptions. Even the validation methods are not direct approach for the computational evaluation. However, we’d like to add discussion why each of them should not be included.
>
> --Include appropriate extension experiments on high-dimensional settings (e.g., small-scale LLMs or vision models)
>
> QUESTION: Lack of empirical support for generalization to "high-dimensional agents"?
>
> RESPONSE: We focus on the EFFICIENT meta-evaluation theory for MINI agent in this paper. The parameter dimension of the synesthetic system does not influence bounds as shown in the theorem 4. Our mini agent dimension is about 200-1000 (The MLP) which is regarded as larger enough to prove its effectiveness comparing with previous works that proved causal ability in random experiments (<3 binary dimension). Whether the scaling law works for computational evaluation in super-larger dimensions space, such as billion scale, are underexplored.
>
> QUESTION: The paper claims the framework …… how parameter dimensions affect the error bounds.
>
> RESPONSE: The data we used to build the true evaluation system is unrelative with the dimension of the agents being input. The dataset is just a mediator to attain the ‘true’ metric following the pointed correlation between those metrics and predictive metrics in RCT datasets at the line 236-239. The input of the evaluation model is the mini agent and condition before deploying the mini agent. For example, our MLP agent’s dimension is about 400-500 in the task to use AI decided alert box to improve the mortality.
>
> QUESTION: For example, Theorem 2 mentions …….. may lead to the curse of dimensionality and overfitting risks.
>
> REPSONSE:
>
> “The curse of dimensionality”: We explored the dimensions from 10-1000 in our experiments. The curse of dimensionality is handled in the learner part. For example, we use CatBoost tree structures (input is the mini agent and condition before deploying the mini agent, output is the evaluation metric) which is a worked prior in our trade scene.
>
> “Overfitting risks”:
>
> The main challenge in the metric prediction model (which we called evaluation model in the paper) is ‘underfitting’ rather than overfitting. It is totally different with the detection. In the detection task, we are confident that there exists a process, such as human’s vision, that can detect the object accurately. After hard searching, we are certainly can find good models for detection. While it is difficult to know whether there exists a process, or how good it can be in a metric prediction task without experiments.
>
> QUESTION: How can the validity ……. with millions of parameters (e.g., LLMs)?
>
> RESPONSE: The theoretical bounds size does not depend on the parameters number of the agents. It only depends on the number of samples, and error tolerance. While the empirical error may be larger or smaller, it depends on whether the evaluation model can utilize the huge information from the parameterized encoded LLM vector in the optimization. We left it in our future works.
>
> --Provide more implementation details.
>
> QUESTION: “Could high dimensional …… be used to verify scalability?”
>
> RESPONSE: The ability to handle the high-dimensional synthetic data depends on the specific learners, including its designed training loss. The sparse regularization strategies have been used in some of our meta-learners, such as Het-Linear. The regularization parameters used in our work is listed in the appendix F. The penalty we used in LogisticRegression is L2. The penalty parameter is C=1.0. Whether to use the penalty parameter depended on the structure of the subject vector space, condition space and proxy space.
>
> QUESTION: “Missing pseudocode for the …… which undermines reproducibility.”
>
> RESPONSE: Our vectorization strategy: flatten the parameters of the mini agent into a vector without any structure. We also publish the code and data for reproduction.
>
> We sincerely hope the additional experiments and the responses can alleviate reviewer’s concerns and improve the ratings.

---

> ### Author Response · Authors · 2025-08-08
>
> Dear Reviewer Mu2b,
>
> Thank you for highlighting key concerns regarding robustness, baselines, the need for additional experiments, and details. We respectfully invite you to review our detailed rebuttal and updated results.
>
> We sincerely hope that our responses and the newly added experiments adequately address your concerns. We believe the contributions of this work justify a more favorable rating, and we would greatly appreciate your reconsideration.
>
> Best regards,
> Authors

---

### Official Review · Reviewer_bzq1 · 2025-07-05

**Clarity:** 2
**Significance:** 3
**Originality:** 2
**Rating:** 3
**Confidence:** 2

**Summary:**

The article introduces a computational evaluation framework for small agents, which focuses on accelerating the evaluation process by constructing evaluation models. The framework is built on the basis of rigorous derivation of upper bounds on generalized evaluation error and causal effect error, and the article also proposes a meta-learning strategy to solve the problem of heterogeneity in the agent space. The article conducts a large number of evaluation experiments for different application scenarios, which proves that its proposed computational evaluation framework achieves certain results in reducing the evaluation error and improving the evaluation speed.

**Questions:**

See above

**Ethical Concerns:**

["NO or VERY MINOR ethics concerns only"]

**Final Justification:**

I suggest the author to learn the Markdown language carefully. This does not sound like a reply from an academic, but rather from an undergraduate paper.

**Limitations:**

See above

**Quality:**

2

**Strengths And Weaknesses:**

Strengths:

The article proposes a computational evaluation framework for causal effect assessment of small agents and theoretically rigorously derives upper bounds for the evaluation error and causal effect error, the article has some novelty and theoretical depth. The article is experimentally validated in 12 real-world scenarios covering multiple domains such as healthcare, simulation, and business, which fully illustrates its excellent generalizability. The article proposes a meta-learning strategy to adapt to the problem of heterogeneity in the agent space, which improves the robustness of the model in different situations.And the model achieves a certain degree of leadership in error reduction and evaluation efficiency, proving its application value.

Weaknesses:

The article's assessment metrics may be relatively homogeneous, focusing mainly on error reduction and assessment time acceleration, and lacking further analysis of other important assessment dimensions, such as fairness, robustness, and interpretability. The theoretical framework for computational assessment proposed in the article is based on the core assumption of the validity of its method, for which further systematic empirical and theoretical analysis is lacking. Whether the strong assumptions on which the theoretical analysis of the method relies are satisfied in real scenarios affects its generalization and generalizability. Moreover, the article does not provide a detailed mechanistic explanation of why the model performs well or fails in complex situations.

---

> ### Author Rebuttal · Authors · 2025-07-30
>
> Thanks for the reviewer’s positive comments. Here are the point-to-point responses.
>
> 1.“The article's assessment …… fairness, robustness, and interpretability.”
>
> We agree that computational evaluation’s performance metrics should be heterogeneous. So, we add/indicate the following experiments to show the fairness, robustness, and interpretability of the proposed evaluation approach:
>
> ---- Fairness: Fairness is a kind of inside attribute of our evaluation model once the IIDE is satisfied. The fairness of evaluation model depended on the independently sampling distribution of mini agents. Specifically, we use Gaussian distribution as prior which is often regarded as fair for potential mini agents.
>
> Furthermore, we add absolute difference on calculus score (sex fairness) as extra fairness metrics of mini agents to address the review’s concern. The computational evaluation for fairness is still worked, the following is the reported evaluation errors for fairness metric:
>
> Evaluation Model | Score Error Difference between Male and Female
>
> Het-Linear | 0.0062
>
> Het-RF | 0.0048
>
> Het-LGBM | 0.0045
>
> Het-XGB | 0.0046
>
> Het-CatBoost | 0.0044
>
> It shows that our evaluation models can still give an accurate sex fairness metric prediction of mini agents with score error lower than 0.01 (Normalized score with mean 0 and std 1).
>
> ---- Robustness: We add an experiment to show the robustness of our proxy module in the evaluation models.
>
> The Gaussian noises strength added into the proxy metrics in the test set are 0.01,0.02,0.03,0.04,0.05:
>
> Task 1: evaluate the mini agents that using individualized alert box to improve the mortality
>
> 1)Gaussian noise on proxy metrics (Het-Linear) | 0 | 0.01| 0.02 | 0.03 | 0.04 | 0.05|
>
> Error of ROC_AUC | 0.01659 | 0.01657 | 0.01679 | 0.01654 | 0.01915 | 0.01663|
>
> 2)Gaussian noise on proxy metrics (Het-MLP) | 0 | 0.01| 0.02 | 0.03 | 0.04 | 0.05|
>
> Error of ROC_AUC | 0.04242 | 0.04478 | 0.04776 | 0.04263 | 0.04308 | 0.04341|
>
> 3)Gaussian noise on proxy metrics (Het-SVR) | 0 | 0.01| 0.02 | 0.03 | 0.04 | 0.05|
>
> Error of ROC_AUC | 0.03555 | 0.03532 | 0.03502 | 0.03429 | 0.03596 | 0.03483|
>
> 4)Gaussian noise on proxy metrics (Het-RF) | 0 | 0.01| 0.02 | 0.03 | 0.04 | 0.05|
>
> Error of ROC_AUC | 0.01998 | 0.01989 | 0.02009 | 0.01993 | 0.02019 | 0.01990|
>
> 5)Gaussian noise on proxy metrics (Het-LGBM) | 0 | 0.01| 0.02 | 0.03 | 0.04 | 0.05|
>
> Error of ROC_AUC | 0.01883 | 0.01871 | 0.01894 | 0.01891 | 0.01915 | 0.01874|
>
> 6)Gaussian noise on proxy metrics (Het-XGB) | 0 | 0.01| 0.02 | 0.03 | 0.04 | 0.05|
>
> Error of ROC_AUC | 0.02055 | 0.02028 | 0.02075 | 0.02049 | 0.02070 | 0.02053|
>
> 7)Gaussian noise on proxy metrics (Het-CatBoost) | 0 | 0.01| 0.02 | 0.03 | 0.04 | 0.05|
>
> Error of ROC_AUC | 0.01745 | 0.0173 | 0.01756 | 0.01747 | 0.01771 | 0.01725|
>
> Task 2: evaluate the mini agents that using individualized buy-in decision to improve the RoI of trade
>
> Gaussian noise on proxy metrics (Het-Linear) | 0 | 0.01| 0.02 | 0.03 | 0.04 | 0.05|
>
> Error of RoI |2.1630| 1.5773| 1.9705| 1.9071| 1.7212| 1.70865|
>
> Form the experiment results, evaluation model performance is not easy to be intervened by the noise proxy metrics. The fluctuate of the performance may be introduced by the randomness.
>
> ----the interpretability of the modules in the computational evaluation model has been introduced in the section 6.3 line 280-295.
>
> 2. “The theoretical framework …… its generalization and generalizability.“
>
> The effectiveness of mentioned assumption IIDE and unbiases has been assessed by the assessment procedures. It shows the generalization of IIDE and unbiased assumption. The assessment table in the appendix shows strong generalization and generalizability of the assumptions in those real scenarios at least. And our bound are not very tight so that it is robust in those cases. Here is additional robustness analysis.
>
> (a)	IIDE assumption
>
> 1)How to satisfy it and why it is useful?
>
> We remark that all the error is hoped as IID by researchers. If it is not IIDE, the best choice is not to test its robustness to increase the confidence of the assumptions but to change the metric prediction model to make the errors unpredictable (boosting algorithms can make it unpredictable for given functions), independently, and randomly. It gives us a SECOND CHANCE to satisfy the assumption if the (s,c,m) is not IID. This is the main difference in theory between our work and other similar works based on the IID assumption.
>
> 2)Empirical non-IIDE case study
>
> We design 3 synthetic experiments to show the bound robustness of the IIDE. The first non-IIDE situation is distribution shifting of errors. The second non-IIDE situation is error dependency (temporal). The third non-IIDE situation is the outliers (black swan events).
> The condition is N(0,1). The mini agent assignment is N(0,1). The noise is N(0,0.1). The evaluation metric function f(c,s,e) is np.clip(0.3 * c + 0.5 * s + e, 0, 1). The evaluation metric predictive model is np.clip(0.3 * c + 0.5 * s, 0, 1). The number of samples is 500.
>
> Here is the robustness test experiment result:
>
> Name |Our Theoretical Bound | IIDE | non-IIDE1 | non-IIDE 2| non-IIDE 3|
>
> Error Bound |0.06484732406369578 | 0.010114041012576054| 0.03348055885517107| 0.039488422074875774| 0.02999711253327391|
>
> Individual Causal Effect Error Bound |0.1096772717167341 |0.00010535280724731346|0.08964805476948194| 0.10440750588976608| 0.08760353747620356|
>
> From the experiments, we can see that even in those non-IID cases, our evaluation error bound, and causal evaluation error bound are still worked.
>
> (b)	IRIS assumption
>
> 1)When to check it?
>
> IRIS assumption is not necessary if the researcher does not need the causal evaluation. All the following discussion is for the causal evaluation rather than pure computational evaluation.
>
> 2)How to satisfy it?
>
> The mini agents are randomly deployed in 11 real scenes. So, the IRIS assumption is always satisfied in our experiments. Before we collect the data, we should also collect the data with randomly deployed mini agents to make the IRIS to be true.
>
> 3)The central role of IRIS assumption for causal property of evaluation model
>
> If the IRIS assumption is violated, we cannot exclude the existence of “hidden common cause which value is IDENTICAL with the mini agent assignment” without new STRONG assumptions. As said as Fisher “it would be impossible to present an exhaustive list of such possible differences appropriate to any kind of experiment, because the uncontrolled causes that may influence the result are always strictly innumerable”. Our further reason to the quote is: there always exists HICC-like unmeasurable confounders, measurable confounders that we do not take into consideration, measurable confounders whose measurement is difficult, or measurable confounders whose measurement costs too much in all kinds of evaluations, which has been demonstrated in the introduction.
>
> 4)Emperical non-IRIS case study
>
> For example, for temporally correlated sampled agents and adversarial-generated agents that violate the IRIS, the similar works can be found in Time-series Deconfounders, and GANITE. as pointed in line 967-972 of section E and line 114-119 of section 3.2. However, those assumptions are regarded as too STRONG for the computational evaluation task with causal guarantee.
>
> Despite the analysis, the bound is also worked for temporally correlated agents and adversarial-generated agents, we recommend using Markov process, and Game theory for detailed analysis. Here we give an empirical result:
>
> The sample number is 500 with 95% confidence. The c is sampled from N(0,1). The e is sampled from N(0,1). For temporally-correlated agents, it is generated by AR(1) to simulate the temporal correlation.
>
> For adversarially-generaed agents, it is generated by alternately pushed the agents towards extreme values (0.05 and 0.95), to simulate the worst-case scenario of adversarial selection.
>
> The results is:
>
> Name |Our Theoretical Bound | Temporally-Correlated Agents | Adversarially-generated Agents |
>
> MSE | 0.003688879454113936 | 0.007143852947639841 | 2.435916193660976e-33 |
>
> Causal Effect MSE | 0.12147229238166103 | 2.1678267454022727e-33 | ~0.0|
>
> From the experiments, even in those non-IRIS cases, our evaluation error bound, and causal evaluation error bound are still worked.
>
> (c)	UNBIASED assumption
>
> 1)How to satisfy it?
>
> We hope the prediction model is unbiased. If it is not, then we can adjust our model and minus the empirical bias from the current data to increase our confidence that it is unbiased.
>
> 2)How robustness it is?
>
> It is trivial to extend the bound in theorem 1 and theorem 4 to prediction model with bias as pointed in the line 159:
>
> Evaluation Error Up Bound: $E_{emp}(\hat{f})+\sqrt{\frac{1}{2n}\log(\frac{1}{\sigma})}-B)$
>
> Causal Effect Error Up Bound: $ 2(E_{emp}(\hat{f})+\sqrt{\frac{1}{2n}\log(\frac{1}{\sigma})}-B))$
>
> Where B is the metric prediction bias $E(\epsilon)$. The minus term does not mean it would be smaller than unbiased cases. Because the bias will also increase the $E_{emp}(\hat{f})$.
>
> For Cauchy-like distribution without bias, it will not be discussed in this paper because it is usually regarded as a special counterexample theoretically rather than a real case in the world.
>
> 3. “Moreover, the article does not provide a detailed mechanistic explanation of why the model performs well or fails in complex situations. “
>
> In our interpretability part in line 280-295 of section 6.3. Taking the trade situations as example, the Shapley value of the proxy metric, subject vector, and condition are about 30-40% respectively to the improve the performance of our evaluation models.
>
> We sincerely hope the additional experiments and the responses can alleviate reviewer’s concerns and improve the ratings.

---

> > ### Comment · Reviewer_bzq1 · 2025-08-04
> >
> > I suggest the author to learn the Markdown language carefully. This does not sound like a reply from an academic, but rather from an undergraduate paper.

---

> ### Author Response · Authors · 2025-08-04
>
> We appreciate the reviewer’s continued engagement with our submission.
>
> > *"This does not sound like a reply from an academic, but rather from an undergraduate paper."*
>
> We respectfully believe that the focus of academic peer review should remain on the **substantive contributions, empirical rigor, and theoretical soundness** of the work rather than the fancy format. The previous response was structured to **directly and thoroughly address all of the reviewer’s scientific concerns**, including:
>
> - Additional experiments on **fairness**, **robustness**, and **interpretability**;
> - Empirical validation under **violated assumptions** (non-IID errors, non-IRIS agents);
> - Mechanistic insight through **Shapley-value-based model interpretation**;
> - Extension of **theoretical bounds** under relaxed conditions.
>
> All these were provided with detailed quantitative results and methodological clarity.
>
> We acknowledge that stylistic preferences vary across reviewers. Nevertheless, the current form of our rebuttal **adequately serves the goal of scientific communication**: enabling reviewers to assess whether their concerns have been resolved and whether the work deserves to be accepted.
>
> We trust that the **technical merit**, **breadth of real-world applications**, and **rigorous empirical support** in both the paper and the rebuttal are sufficient to warrant a **higher rating**.
>
> We sincerely encourage the reviewer to **reconsider the numerical scores** based on the substance of our responses and the overall contribution of this work to computational evaluation under causal guarantees.
>
> Should there be any additional concerns or unresolved issues, we would be pleased to engage further and provide clarifications as needed.

---

> > ### Comment · Reviewer_bzq1 · 2025-08-04
> >
> > I hope the author's response to this is written by himself, rather than generated by a large model. Unfortunately, the author still used a large model to respond.

---

> ### Author Response · Authors · 2025-08-04
>
> We appreciate the reviewer’s suggestions. The large model was utilized exclusively to resolve format-related issues, and not for addressing substantive points. If the reviewer have any further suggesstions that could strengthen our work, we would be glad to engage in discussion or consider them in future developments.

---

### Note · Authors · 2025-08-14

We thank all reviewers for their valuable insights and constructive feedback. We are encouraged that several reviewers recognized the strengths of our work:

- **Clearly addressed problem:** Evaluating many (or infinite) mini agents via randomized experiments to get the (conditional) causal effect from mini agents to evaluation metrics is prohibitively time-consuming.
- **Promising idea:** Modeling the evaluation process with an *evaluation model* enables a speedup of ~10³–10⁷.
- **Theoretical advantage:** Compared with existed IID-assumption-based causal evaluation, our theory with weaker IIDE assumptions offers a *SECOND* opportunity to satisfy assumptions by adjusting evaluation models after data generation (first opportunity).
- **Practical evidence:** Our modules show strong effectiveness (low error) and efficiency (short time) in diverse tasks: improving mortality via invidualized alert/drug-withdrawal, raising grades via individualized class assignment, increasing trading ROI, and enhancing user conversion for advertising/insurance.

**Addressing Common Concerns**
- **Fairness, robustness, and interpretability of the evaluation model:** Additional experiments provide strong evidence supporting these properties.
- **Assumption Robustness:** Detailed robustness analysis and new empirical tests were added.
- **Baseline selection:** No existing methods are directly designed for computational evaluation with causal guarantees as we know; nonetheless, we included some model-free validation approaches as baselines and SOTA models (RF, XGBoost, CatBoost) for tabular data as part of our learners.

**Planned Revisions**
1. Correct typos.
2. Expand discussion on assumption robustness.
3. A more detailed discussion about the baselines.

We note that the key issues raised by two reviewers Mu2b and gSDJ were directly addressed with new results, but they did not participate in the post-rebuttal discussion. Given these substantial revisions and the lack of further engagement from those reviewers, we believe the updated quality and contributions should be assessed based on the current version, which we consider to meet the standard for acceptance and to offer meaningful impact for the **EVALUATION** community.

---

### Decision · Program_Chairs · 2025-09-17

**Decision:**

Reject

**Comment:**

The manuscript proposes an evaluation framework for small agents, which accelerates evaluation via evaluation models. The framework is evaluated on a set of 12 real-world scenarios, where it exhibits large empirical gains in error reduction and acceleration.
On the positive side, reviewers largely agree that the proposed method is novel and that the article has theoretical depth. However, reviewers had doubts about the appropriateness of assuming independently identically distributed errors. It was also pointed out that the baselines were not particularly strong and that the framework should have been compared to newer methods like deep causal networks or counterfactual modeling. The claim that the framework is applicable to high-dimensional agents is not well supported, as experiments are carried out on low-dimensional spaces. The authors addressed some of the raised concerns, however the rebuttal was still regarded as not meeting the requirements for acceptance. Specifically, a lack of theoretical analysis on the framework’s robustness and parameters sensitivity, as well as the absence of a dedicated related work section were cited as reasons to maintain the reject ratings.